# IS SOFTMAX LOSS ALL YOU NEED? A PRINCIPLED ANALYSIS OF SOFTMAX LOSS AND ITS VARIANTS

## ABSTRACT

**The Softmax Loss** is one of the most widely employed surrogate objectives for classification and ranking, owing to its elegant algebraic structure, intuitive probabilistic interpretation, and consistently strong empirical performance. To elucidate its theoretical properties, recent works have introduced the **Fenchel–Young** framework, situating Softmax loss as a canonical instance within a broad family of surrogate losses. This framework not only clarifies the origins of its favorable properties, but also unifies it with alternatives such as Sparsemax and $\alpha$-Entmax under a principled theoretical foundation. Concurrently, another line of research has addressed on the challenge of scalability: when the number of classes is exceedingly large, computations of the partition function become prohibitively expensive. Numerous approximation strategies have thus been proposed to retain the benefits of the exact objective while improving efficiency. However, their theoretical fidelity remains unclear, and practical adoption often relies on heuristics or exhaustive search.

Building on these two perspectives, we present a principled investigation of the **Softmax-family** losses, encompassing both statistical and computational aspects. Within the Fenchel–Young framework, we examine whether different surrogates satisfy consistency with classification and ranking metrics, and analyze their gradient dynamics to reveal distinct convergence behaviors. For approximate Softmax methods, we introduce a systematic bias–variance decomposition that provides convergence guarantees. We further derive a per-epoch complexity analysis across the entire family, highlighting explicit trade-offs between accuracy and efficiency. Finally, extensive experiments on a representative recommendation task corroborate our theoretical findings, demonstrating a strong alignment between consistency, convergence, and empirical performance. Together, these results establish a principled foundation and offer practical guidance for loss selections in large-class machine learning applications.

## 1 INTRODUCTION

The **Softmax cross-entropy loss** has become one of the most widely adopted objectives in modern machine learning, underpinning state-of-the-art results in domains such as language modeling, machine translation, computer vision, and recommender systems (Mikolov et al., 2013a; Sutskever et al., 2014; He et al., 2016). Its widespread adoption can be attributed to several appealing properties: a smooth and differentiable formulation well-suited for gradient-based optimization, a probabilistic interpretation with geometric intuition, and strong alignment with classification and ranking objectives. These features have established it as the de facto standard across diverse architectures and optimizers. Representative examples include the Transformer (Vaswani et al., 2017) and GPT models (Brown et al., 2020), whose final Softmax layer, combined with cross-entropy loss, forms a fundamental component of the overall objective.

From a theoretical standpoint, these favorable properties can be unified under the **Fenchel–Young (F-Y)** loss framework (Blondel et al., 2019), which recovers Softmax cross-entropy as a particular instance associated with negative Shannon entropy. This framework not only explains the effectiveness of Softmax but also situates it within a broader family of geometrically grounded surrogates, including Sparsemax (Martins & Astudillo, 2016) and $\alpha$-Entmax (Peters et al., 2019), each encoding different inductive biases.

Table 1: Exact Softmax-family properties.

| Property | Softmax | Sparsemax | $\alpha$-Entmax | Rankmax |
|---|---|---|---|---|
| Consistency? | ✓ | ✓ | ✓ | ✓ |
| Ordering? | SOP | WOP | WOP | WOP |
| Smoothness ($\|J\|_2$) | 1/2 | 1 | 1 | 1 |
| Per-epoch Complexity | $\Theta(NC)$ | $\Theta(NC \log C)$ | $\Theta(NC \log C + N|P|)$ | $\Theta(NC)$ |

Table 2: Approximate method[1] properties.

| Property | SSM | NCE | HSM | RG |
|---|---|---|---|---|
| Consistency? | Asymptotic | Asymptotic | $\times$ | ✓ |
| Smoothness ($\|J\|_2$) | 1/2 | 1/4 | 1/2 | – |
| Per-epoch Complexity | $\Theta(Nk)$ | $\Theta(Nk)$ | $\Theta(N \log C)$ | – |
| Bias (asymptotic) | 0 | $(1 + k) \, \mathrm{JS}_\tau(P_s \| Q)$ | $\mathrm{KL}(P_s \| P_{\mathrm{HSM}})$ | $O(\|s\|^3)$ |
| Bias (curvature) | $\frac{1}{2k} \chi^2(Q \| P_s)$ | 0 | 0 | 0 |
| Variance | $\frac{1}{k} \chi^2(P_s \| Q)$ | $\frac{k}{(1+k)^2} \chi^2(P_s \| Q)$ | 0 | 0 |

While this theoretical perspective provides a unifying foundation, another major challenge arises from **computational efficiency**. When the number of classes $C$ is extremely large, computing the log-partition term $\log \sum_{j=1}^{C} e^{s_j}$ becomes a prohibitive bottleneck. This has motivated a wide range of approximate Softmax methods, such as Noise Contrastive Estimation (NCE) (Gutmann & Hyvärinen, 2010) and Sampled Softmax (Jean et al., 2015), which trade exactness for tractability.

Although methods from both directions (referred to as **'Softmax-family'**) have been extensively studied, they have largely been progressed in isolation. This fragmented perspective leaves open critical questions: to what extent do these surrogates preserve the theoretical guarantees of Softmax Loss? What computational trade-offs do they entail, and how do they affect optimization dynamics and empirical performance? Addressing these questions requires a common theoretical foundation that enables rigorous comparisons across the entire Softmax-family of losses, not only in terms of gradients or empirical efficiency, but also through the geometry of the surrogate loss landscapes they induce. Without such a foundation, the choice of surrogate losses remains ad hoc and heuristic-driven.

To this end, this work provides a principled study of Softmax-family losses from both statistical and computational perspectives. Our contributions are as follows:

- We establish the **consistency properties** of different F-Y surrogates, determining whether they satisfy fundamental alignment with classification and ranking metrics.

- We analyze the **gradient dynamics** of Softmax-family losses, characterizing their convergence behaviors and revealing differences in optimization efficiency.

- For approximate Softmax methods, we propose a systematic **bias–variance decomposition** that yields convergence guarantees across training scenarios.

- We provide a comparative analysis of the **per-epoch computational complexity** for the entire Softmax family, clarifying trade-offs between statistical accuracy and efficiency.

- Finally, we validate our theoretical findings with **extensive experiments** on a representative recommendation task, demonstrating that consistency, convergence, and computational analysis translate directly into empirical performance.

---

[1]Definition of following methods are in Appendix A.5.

Together, all the theoretical results are concluded in Table 1, 2, which deliver a unified theoretical foundation and offer practical guidelines for selecting surrogate losses in large-class learning scenarios, bridging the gap between theoretical analysis and real-world efficiency.

## 2 FOUNDATIONS OF LEARNING FRAMEWORKS AND LOSSES

### 2.1 PRELIMINARIES

We establish a general supervised learning framework that unifies multi-class/multi-label classification and ranking. Let $\mathcal{X}$ be the input feature space. For any input $\boldsymbol{x} \in \mathcal{X}$, the goal is to predict a relevance distribution over a fixed set of $C$ candidates. The ground-truth label space is defined as $\mathcal{Y} = \{0, 1\}^C$. Label $\boldsymbol{y} \in \mathcal{Y}$ is a binary vector where $y_i = 1$ indicates that $i$ is relevant to input $\boldsymbol{x}$, and $y_i = 0$ indicates not. This representation naturally captures different tasks:

- **Multi-class Classification:** The label $\boldsymbol{y}$ is a one-hot vector, where exactly one element is 1, i.e., $\sum_{i=1}^{C} y_i = 1$.
- **Multi-label Classification & Ranking**[2]: The label $\boldsymbol{y}$ is a multi-hot vector, where one or more elements can be non-zero. The set of relevant items is given by the support of $\boldsymbol{y}$, denoted $\text{supp}(\boldsymbol{y}) = \{i \in \{1, \ldots, C\} : y_i = 1\}$.

We assume training data $(\boldsymbol{x}, \boldsymbol{y})$ is drawn i.i.d. from an unknown distribution over $\mathcal{X} \times \mathcal{Y}$. The model is a parameterized function $f : \mathcal{X} \to \mathbb{R}^C$ with parameters $\theta$, which produces a vector of scores (logits) $\boldsymbol{s} = f(\boldsymbol{x}; \theta) \in \mathbb{R}^C$ for any given input. To facilitate training within a probabilistic framework, the score vector $\boldsymbol{s}$ is subsequently normalized into a probability distribution $\hat{\boldsymbol{p}} \in \Delta^C$, where $\Delta^C = \{\boldsymbol{p} \in \mathbb{R}_+^C : \sum_{i=1}^{C} p_i = 1\}$ is the probability simplex. This mapping is a key component of the surrogate loss function, with the Softmax function being the most prevalent choice.

### 2.2 EVALUATION METRICS

The ultimate objective in supervised learning is to optimize performance with respect to task-specific evaluation metrics. For classification tasks, common choices include Top-$k$ Accuracy and Precision@$k$, while ranking tasks are often evaluated by position-sensitive measures such as NDCG. These metrics, however, are inherently discrete and non-differentiable, and thus cannot be directly optimized using gradient-based methods. Consequently, training relies on smooth, differentiable surrogate loss functions. A central theoretical challenge is to guarantee that minimizing a surrogate loss consistently improves the true task metrics of interest. Formal definitions, notations, and further discussions of these metrics are provided in Appendix A.1.

### 2.3 SOFTMAX LOSS

The softmax mapping is defined as

$$\hat{p}_{\text{SM}}(\boldsymbol{s})_i = \frac{\exp(s_i)}{\sum_{j=1}^{C} \exp(s_j)} \ \in \ \Delta^C, \tag{1}$$

Correspondingly, the softmax cross-entropy (negative log-likelihood) surrogate loss[3] is:

$$\mathcal{L}_{\text{SM}}(\boldsymbol{y}, \boldsymbol{s}) = -\sum_{i:y_i=1} \log \hat{p}_{\text{SM}}(\boldsymbol{s})_i = -\sum_{i:y_i=1} \log \frac{\exp(s_i)}{\sum_{j=1}^{C} \exp(s_j)}, \tag{2}$$

This loss corresponds to the maximum likelihood estimator (MLE) under a categorical distribution, rendering it a statistically principled and widely adopted choice. Besides, the loss also

---

[2]Our discussion primarily centers on ranking scenarios with implicit feedback (i.e. labels are binarized), a setting commonly encountered in search and recommendation tasks.

[3]Here we merge the softmax loss form in terms of multi-class and multi-label/ranking, whose consistency properties (Top-$k$ calibration, DCG-consistency) are satisfied respectively.

enjoys classification-calibration(Zhang, 2004a), Top-$k$ calibration(Lapin et al., 2016), and DCG-consistency[4](Ravikumar et al., 2011). These theoretical guarantees ensure Bayes-optimal prediction under risk minimization, thereby explaining the widespread adoption and empirical robustness of the softmax loss across both classification and ranking tasks.

### 2.4 FENCHEL-YOUNG FRAMEWORK AND BREGMAN DIVERGENCE

The Fenchel–Young (F-Y) loss framework (Blondel et al., 2019) provides a principled and unified approach to constructing convex, classification-calibrated loss functions from a chosen convex regularizer $\Omega$. This framework generalizes familiar objectives: for instance, selecting the negative Shannon entropy as $\Omega$ recovers Softmax Loss in Eq. (2). By varying the regularizer, one obtains a rich family of surrogates with distinct inductive biases. Notable examples include Sparsemax (Martins & Astudillo, 2016) and the $\alpha$-Entmax family (Peters et al., 2019). Formal definitions are provided in Appendix A.2.

**Bregman divergence** (Bregman, 1967) plays a central role in characterizing the statistical consistency of learning algorithms. A key result is that if a surrogate loss can be expressed as a Bregman divergence, then its Bayes-optimal predictor coincides with the true posterior distribution (Reid & Williamson, 2011; Blondel et al., 2019). This property establishes a rigorous theoretical foundation for proving that minimizing the surrogate leads to convergence toward the data-generating distribution. Formal definitions and further details on Bregman divergences are provided in Appendix A.3.

**Mirror Descent** It's noteworthy that F-Y framework naturally aligns with the geometry of Mirror Descent (MD)Nemirovsky & Yudin (1983), which can be understood as supervised analogues of mirror-space optimality. A more detailed discussion is provided in Appendix A.4.

### 2.5 COMPUTATIONAL BOTTLENECK AND APPROXIMATION

A central computational challenge of the Softmax cross-entropy lies in the evaluation of the partition function,

$$\log Z(\boldsymbol{s}) = \log \sum_{j=1}^{C} \exp(s_j), \tag{3}$$

which requires summation over all $C$ classes. When $C$ is on the order of millions, as in extreme classification problems, evaluating $\log Z(\boldsymbol{s})$ and its gradient constitutes the principal bottleneck. To mitigate this issue, a variety of approximation strategies have been proposed. Sampling-based approaches include NCE(Gutmann & Hyvärinen, 2010) and Sampled Softmax (Jean et al., 2015). While structural methods such as Hierarchical Softmax (HSM) (Morin & Bengio, 2005), and analytic approximations such as Taylor expansions (RG)(Pu et al., 2025) approximate the log-partition directly. Detailed discussions of these methods are provided in Appendix A.5.

## 3 MAIN RESULTS

### 3.1 CONSISTENCY ANALYSIS OF FENCHEL-YOUNG LOSSES

#### 3.1.1 RELATION BETWEEN F-Y LOSSES AND BREGMAN DIVERGENCES

For F-Y losses, the representation of the loss as a Bregman divergence is not guaranteed for arbitrary regularizers. A direct equivalence is obtained when $\Omega$ is a **Legendre-type** function, i.e., strictly convex with a gradient mapping $\nabla\Omega$ that is a bijection from the interior of its domain to $\mathbb{R}^C$. Under this condition, Blondel et al. (2019) established the following identity:

**Proposition 3.1** (Blondel et al. (2019)). *If the regularizer $\Omega$ is Legendre-type, the Fenchel–Young loss $L_\Omega(\boldsymbol{s}, \boldsymbol{y})$ is equivalent to the Bregman divergence $D_\Omega$:*

$$L_\Omega(\boldsymbol{y}, \boldsymbol{s}) = D_\Omega(\boldsymbol{y}, \hat{\boldsymbol{p}}(\boldsymbol{s})). \tag{4}$$

---

[4]Under binarized assumption.

The Softmax loss is the canonical example of this principle, since its regularizer, the negative Shannon entropy, is Legendre-type. Consequently, the Bayes-optimal prediction $\boldsymbol{p}^*$ is guaranteed to coincide with the true posterior distribution $\boldsymbol{\eta}$.

To translate posterior matching into metric consistency, the inverse mapping from $\boldsymbol{p}^*$ back to the optimal scores $\boldsymbol{s}^*(\boldsymbol{x})$ must preserve task-relevant orderings. These requirements are formalized by the *inverse top-k preserving* property for Top-$k$ calibration (Lapin et al., 2016; Yang & Koyejo, 2020) and the *inverse order-preserving* property for DCG consistency (Ravikumar et al., 2011). The strictly monotonic mapping induced by the Softmax function satisfies both conditions, thereby ensuring broad consistency.

**Proposition 3.2.** *The Softmax loss is classification-calibrated, Top-k calibrated for all k, and DCG-consistent.*

Notably, sampling-based approximate variants of Softmax inherit these properties only asymptotically. SSM and NCE yield asymptotically unbiased gradients and are Softmax–MLE consistent (Gutmann & Hyvärinen, 2010) as the number of negative samples $k$ increases. For non-sampling approximations, the consistency of HSM and RG has been systematically analyzed in (Wydmuch et al., 2018; Pu et al., 2025), showing that HSM does not preserve consistency whereas RG does.

### 3.1.2 CONSISTENCY FOR SPARSE F-Y LOSSES

In contrast to Softmax, sparse F-Y losses such as Sparsemax and $\alpha$-Entmax arise from non-Legendre regularizers, namely the squared $\ell_2$-norm and the Tsallis entropy. Consequently, these losses do not coincide with their associated Bregman divergence but only upper bound it (Blondel et al., 2019). This lack of equivalence implies that the consistency guarantees of the Softmax loss cannot be directly transferred to these sparse alternatives.

More fundamentally, sparse F-Y losses induce sparsity by assigning zero probability to low-scoring classes for many inputs. This inductive bias de-emphasizes the reproduction of small non-zero probability masses. Nevertheless, what matters for calibration in classification and ranking is the preservation of orderings: the Bayes-optimal solutions of Sparsemax and $\alpha$-Entmax exactly preserve the relevant order structure. As a result, they retain Top-$k$ calibration and DCG consistency.

**Proposition 3.3.** $\forall k > 1$, *Sparsemax and $\alpha$-Entmax are Top-k calibrated and DCG-consistent.*

*Proof.* See Appendix B.1. $\square$

### 3.1.3 HIDDEN DANGERS FOR SPARSE ALIGNMENTS: INSUFFICIENT ORDER PRESERVATION

While both dense (Softmax) and sparse F-Y losses are Top-$k$ calibrated, their optimization behavior diverges due to fundamental geometric properties of the **prediction mapping**. For Softmax, the inverse is available in closed form ($s_i = \log \hat{p}_i + c$). By contrast, sparse mappings lack a simple inverse and are not bijective: they are not injective, meaning distinct score vectors $\boldsymbol{s}$ can be mapped to the same probability vector $\hat{\boldsymbol{p}}$. This induces a **lossy compression** of the input scores, discarding information about the ordering of lower-ranked logits.

This compressive property is the root of the differing training dynamics. To analyze it rigorously, we find that the concept of **order preservation** is not monolithic, but rather splits into two fundamentally different regimes:

**Definition 3.4** (Order Preservation). Let $\hat{p} : \mathbb{R}^C \to \Delta_C$ be a prediction mapping.

- **Strictly order preserving (SOP).** $\hat{p}$ is SOP if for any $i \neq j$, $s_i > s_j \iff \hat{p}_i(\boldsymbol{s}) > \hat{p}_j(\boldsymbol{s})$ holds for all $\boldsymbol{s} \in \mathbb{R}^C$.

- **Weakly order preserving (WOP).** $\hat{p}$ is WOP if for any $i \neq j$, $s_i > s_j \Rightarrow \hat{p}_i(\boldsymbol{s}) \geq \hat{p}_j(\boldsymbol{s})$ holds for all $\boldsymbol{s}$, and there exist $i \neq j$ with $s_i > s_j$ yet $\hat{p}_i(\boldsymbol{s}) = \hat{p}_j(\boldsymbol{s})$.

As for Softmax, $\hat{p}_i(\boldsymbol{s}) = \exp(s_i)/\sum_j \exp(s_j)$ is strictly increasing in each coordinate and even strictly Schur-isotone, hence $s_i > s_j \iff \hat{p}_i(\boldsymbol{s}) > \hat{p}_j(\boldsymbol{s})$ for all $i \neq j$. Therefore, Softmax is SOP and admits no ties except on the logit-equality hyperplanes $s_i = s_j$.

For Sparsemax, the prediction takes the "shifted-thresholded" form

$$\hat{p}_i(\boldsymbol{s}) \;=\; \max\{\, s_i - \tau(\boldsymbol{s}),\, 0 \,\}, \qquad \tau(\boldsymbol{s}) \text{ chosen s.t. } \sum_i \hat{p}_i(\boldsymbol{s}) = 1. \qquad (5)$$

Let $\mathcal{P}(\boldsymbol{s}) := \{i : \hat{p}_i(\boldsymbol{s}) > 0\}$ denote the support set. Then for $i \in \mathcal{P}(\boldsymbol{s})$ and $j \notin \mathcal{P}(\boldsymbol{s})$ we have $s_i - \tau(\boldsymbol{s}) > 0 \geq s_j - \tau(\boldsymbol{s})$, hence $\hat{p}_i(\boldsymbol{s}) > \hat{p}_j(\boldsymbol{s}) = 0$. However, whenever two logits straddle the threshold with $\tau(\boldsymbol{s}) \geq s_i > s_j$, we obtain $\hat{p}_i(\boldsymbol{s}) = \hat{p}_j(\boldsymbol{s}) = 0$, yielding a tie. Analogous piecewise-thresholding holds for $\alpha$-Entmax (with a different nonlinearity but the same support-inducing mechanism). Thus sparse methods are order preserving in the weak sense, yet admit nontrivial plateaus with $\hat{p}_i = \hat{p}_j$ for $s_i > s_j$ whenever both lie at or below the active threshold.

**Proposition 3.5** (Sparse methods are WOP, not SOP). *Sparsemax and $\alpha$-Entmax are weakly order preserving but not strictly order preserving.*

*Proof sketch.* Monotonicity without inversions follows from the thresholding structure above. Non-strictness is witnessed by any $s$ with $\tau(\boldsymbol{s}) \geq s_i > s_j$, which yields $\hat{p}_i(\boldsymbol{s}) = \hat{p}_j(\boldsymbol{s}) = 0$; hence ties occur while logits are strictly ordered. $\square$

**Theorem 3.6** (WOP is sufficient for Calibration). *Let $\mathcal{L}_\Omega$ be a F-Y loss whose prediction mapping $\hat{\boldsymbol{p}}(\boldsymbol{s}) = \nabla\Omega^*(\boldsymbol{s})$ is WOP. Then, $\mathcal{L}_\Omega$ is Top-$k$ calibrated and DCG-consistent.*

*Proof sketch.* Similar to the proof of Prop.3.3 in Appendix.B.1. $\square$

The optimization dynamics of F-Y losses are determined by the structure of its Hessian, $H(\boldsymbol{s}) = \nabla_{\boldsymbol{s}}^2 \ell(\boldsymbol{s})$, which is equivalent to the Jacobian of the prediction map, $J(\boldsymbol{s}) = \nabla_{\boldsymbol{s}}\hat{\boldsymbol{p}}(\boldsymbol{s})$. We analyze this structure for both SOP and WOP mappings.

**Dense/SOP (Softmax).** The Jacobian of the Softmax function is given by $J_{\mathrm{sm}}(\boldsymbol{s}) = \mathrm{diag}(\hat{\boldsymbol{p}}(\boldsymbol{s})) - \hat{\boldsymbol{p}}(\boldsymbol{s})\hat{\boldsymbol{p}}(\boldsymbol{s})^\top$. This matrix is symmetric, positive semidefinite, and has a rank of exactly $C - 1$. Its null space is one-dimensional, spanned by the all-ones vector $\mathbf{1}$, i.e., $\ker(J_{\mathrm{sm}}) = \mathrm{span}(\mathbf{1})$, which corresponds to the shift-invariance of the Softmax function. Consequently, the Hessian provides well-conditioned curvature on the tangent space of the simplex, leading to a smooth, convex optimization landscape amenable to gradient-based methods.

**Sparse/WOP (Sparsemax/Entmax).** Conversely, the Jacobian of a WOP mapping is characterized by rank deficiency . The Jacobian $J_{\mathrm{sp}}(\boldsymbol{s})$ is piecewise constant and block-structured. For a fixed support $\mathcal{P}(\boldsymbol{s})$ with size $m := |\mathcal{P}(\boldsymbol{s})|$, the Sparsemax block reads[5]

$$J_{\mathrm{sp}}(\boldsymbol{s})\big|_{\mathcal{P}\times\mathcal{P}} \;=\; I_m - \tfrac{1}{m}\mathbf{1}\mathbf{1}^\top, \qquad J_{\mathrm{sp}}(\boldsymbol{s})\big|_{\mathcal{P}^c\times\mathbb{R}^C} \;=\; \mathbf{0}, \qquad (6)$$

so that $\mathrm{rank}(J_{\mathrm{sp}}) \leq m-1$. Consequently, the Hessian $H_{\mathrm{sp}} = J_{\mathrm{sp}}$ admits large null spaces, yielding extended flat subspaces (zero curvature). Moreover, while $v^\top H_{\mathrm{sp}} v = 0$ for any $v \in \ker(H_{\mathrm{sp}})$, the first-order directional derivative $\langle \nabla_{\boldsymbol{s}}\ell, v \rangle$ vanish for directions supported entirely on off-support coordinates whose gradient components are zero[6], which explains the lack of learning signal for many negatives.

In conclusion, sparse methods preserve the ordering requirements for Top-$k$ calibration and DCG consistency at the decision-level, but their WOP property induces (i) vanishing gradients off-support, (ii) large flat subspaces from Jacobian rank deficiency, all of which hinder optimization in practice. In contrast, SOP losses supply ubiquitous competitive gradients on the simplex tangent space, leading to more stable and efficient training. We provide formal theorems and detailed discussions of the drawbacks of WOP property in Appendix.B.2.

---

[5]For $\alpha$-Entmax, an analogous block-sparse structure holds with rank at most $m-1$, while the within-support coefficients depend on $\alpha$. Detailed in Appendix.B.4.

[6]Visualizations can be found in Appendix.C.5.

### 3.1.4 WOP SPARSE ALTERNATIVE: RANKMAX

A further member of the sparse F-Y family is **Rankmax** (Kong et al., 2020). Rankmax explicitly conditions the support on the score of the ground-truth class, and is also weakly order preserving: strict logit inequalities may collapse into ties but are never inverted. In contrast to Sparsemax and $\alpha$-Entmax, which zero out off-support classes and may suffer from widespread vanishing gradients of hard negatives, Rankmax employs a ground-truth–centered threshold. This design achieves a targeted trade-off: it produces distributions that are sparser and more focused than dense Softmax, while mitigating the shortcomings of purely threshold-based sparse mappings that may discard informative hard negatives. A detailed theoretical analysis is deferred to Appendix B.3.

## 3.2 CONVERGENCE RATE OF SOFTMAX-FAMILY

Building on the analysis of Jacobian dynamics, we further analyze how the spectral norms of their Jacobians characterize the smoothness of the objective.

It is crucial to clarify that while deep neural networks are inherently *non-convex* w.r.t. the full parameter set $\theta$, $\mathcal{L}(\boldsymbol{y}, \boldsymbol{s})$ are convex functions w.r.t. the logits $\boldsymbol{s}$ by construction. Consequently, the projection layer forms a strongly convex subproblem with $l_2$ regularization, whose smoothness plays a significant role in the optimization of the entire network, as it directly modulates the gradients back-propagated to the feature extractor $s_\theta(\cdot)$. Formally, using the chain rule $\nabla_\theta \mathcal{L} = J_{\theta|\mathbf{s}}^\top \nabla_{\mathbf{s}} \mathcal{L}_{\text{Head}}$, the gradient magnitude w.r.t. feature parameters is upper-bounded by:

$$\|\nabla_\theta \mathcal{L}\| \leq \|J_{\theta|\mathbf{s}}^\top\| \cdot \|\nabla_{\mathbf{s}} \mathcal{L}_{\text{Head}}\| \leq \|J_{\theta|\mathbf{s}}^\top\| \cdot L_{\text{Head}} \cdot \|\mathbf{s} - \mathbf{s}^*\|, \tag{7}$$

where $L_{\text{Head}}$ denotes the smoothness constant of the prediction head. This implies that lower $L_{\text{Head}}$ better scales and shapes the gradient signal received by all earlier layers.

The detailed derivations are deferred to Appendix B.4, which yields the following ordering of linear convergence factors:

$$\rho_{\text{NCE}} < \rho_{\text{SSM}} = \rho_{\text{SM}} = \rho_{\text{HSM}} < \rho_{\alpha-\text{Entmax}} = \rho_{\text{Sparsemax}} = \rho_{\text{Rankmax}}. \tag{8}$$

It is worth noting, however, that although sampling-based objectives admit equal or even smaller Jacobian spectral norms than full-score losses, their practical convergence rates are further affected by the bias introduced by sampling (Appendix B.5). Consequently, they do not guarantee universally equal or faster convergence, and empirical tuning remains necessary in practice.

## 3.3 BIAS–VARIANCE DECOMPOSITION FOR SOFTMAX APPROXIMATIONS

The computational bottleneck of Softmax Loss lies in evaluating the partition function $\log Z(\boldsymbol{s}) = \log \sum_{j=1}^C \exp(s_j)$ (Section 2.5). A rich line of work has proposed approximation strategies (Appendix A.5), typically studied in isolation. We show that these diverse methods can be subsumed under a unified F-Y risk framework. Given a convex potential $\Omega$, the surrogate loss is

$$\ell_\Omega(\boldsymbol{y}, \boldsymbol{s}; \xi) = \Omega^*(\boldsymbol{s}; \xi) + \Omega(\boldsymbol{y}) - \langle \boldsymbol{s}, \boldsymbol{y} \rangle, \tag{9}$$

with expected risk

$$R_\Omega(\theta; \xi) = \mathbb{E}_{(x,y)\sim\mathcal{D}}\big[\ell_\Omega(\boldsymbol{y}, \boldsymbol{s}_\theta(x); \xi)\big]. \tag{10}$$

Here $\xi$ denotes the approximation mechanism, encapsulating all scheme-specific variables (e.g., proposal distribution $Q$, sampling size $k$). Deterministic surrogates (e.g., Softmax, Sparsemax, $alpha$-Entmax) correspond to degenerate $\xi$ without randomness, whereas sampling-based approximations correspond to non-degenerate $\xi$. We keep the definition of $\ell_\Omega$ unchanged and encode all differences through $\xi$.

Taking the exact Softmax risk as reference:

$$R_\star(\theta) := R_{\Omega_{\text{SM}}}(\theta), \qquad \Omega_\star^*(\boldsymbol{s}) = \log \sum_{j=1}^C e^{s_j}, \tag{11}$$

the deviation is

$$\Delta R(\theta;\xi) = R_\Omega(\theta;\xi) - R_\star(\theta), \tag{12}$$

which admits the decomposition

$$\Delta R(\theta;\xi) = \underbrace{\mathbb{E}_\xi\big[R_\Omega(\theta;\xi)\big] - R_\star(\theta)}_{\text{Bias}} + \underbrace{\big(R_\Omega(\theta;\xi) - \mathbb{E}_\xi[R_\Omega(\theta;\xi)]\big)}_{\text{Stochastic noise}},$$

$$\mathbb{E}_\xi\big[\Delta R(\theta;\xi)^2\big] = \text{Bias}^2 + \underbrace{\mathbb{E}_\xi\big[(\text{Stochastic noise})^2\big]}_{\text{Variance}}. \tag{13}$$

This decomposition highlights two distinct effects of approximation:

- **Bias.** Quantifies the systematic deviation between the approximate and exact Softmax risk. It determines (i) whether approximation is effective and preserves consistency guarantees, and (ii) how bias magnitude influences optimization at the loss level.

- **Variance.** Captures stochastic fluctuations induced by sampling. It vanishes for deterministic approximations, but may significantly perturb training dynamics for sampling methods.

By disentangling these two factors, the decomposition provides a principled lens for rigorously comparing existing approximations and understanding their optimization behavior.

### 3.3.1 DELTA-METHOD APPROXIMATION

To analyze the impact of approximations, we employ $\Delta$-method, a classical tool in asymptotic statistics. The detailed setup and analysis can be found in Appendix.B.5. We summarize the results for the approximation schemes of Appendix A.5 in Tab. 3.

Table 3: Bias–Variance characterization of different surrogates relative to softmax cross-entropy. Here $k$ denotes the number of negative samples used in sampling-based methods.

| **Surrogate** | $\text{Bias}^{\text{asym}}$ | $\text{Bias}^{\text{curv}}$ | $\text{Var}$ |
|---|---|---|---|
| Softmax (ref) | 0 | 0 | 0 |
| SSM-Simple | $\log \frac{e^{s_y} + k\mathbb{E}_Q[e^{s_{y'}}]}{\sum_i e^{s_{y_i}}}$ | $-\frac{k}{2(\hat{\Omega}^*)^2}\text{Var}_Q(e^{s_{y'}})$ | $\frac{k}{(\hat{\Omega}^*)^2}\text{Var}_Q(e^{s_{y'}})$ |
| SSM | 0 | $-\frac{1}{2k}\chi^2(P_{\boldsymbol{s}}\|Q)$ | $\frac{1}{k}\chi^2(P_{\boldsymbol{s}}\|Q)$ |
| NCE | $(1+k)\,\text{JS}_\tau(P_{\boldsymbol{s}}\|Q)$ | 0 | $\frac{k}{(1+k)^2}\chi^2(P_{\boldsymbol{s}}\|Q)$ |
| HSM | $\text{KL}(P_{\boldsymbol{s}}\|P_{\text{HSM}})$ | 0 | 0 |
| RG | $O(\|\boldsymbol{s}\|^3)$ | 0 | 0 |

### 3.4 TRAINING COMPLEXITY ANALYSIS

We report *per-epoch* asymptotic costs for the classification head of all Softmax-family losses(excluding the backbone). Let $N$ be the number of training examples per epoch, $C$ the number of classes, $k$ the number of sampled negatives, and $\mathcal{P} = |\text{supp}(\hat{\boldsymbol{p}})|$ the active support for sparse heads, as summarized in Table 4.

## 4 EXPERIMENTS

To complement our theoretical analysis, we identify the following key questions that require systematic experimental validation, which guides the design of our empirical study.

**Q1.** How do sparse alternatives perform relative to Softmax (SOP) on real-world classification and ranking metrics, given their shared consistency guarantees but differing convergence dynamics?

**Q2.** How well do the theoretical bias–variance decompositions align with empirical training behavior of sampling-based methods?

Table 4: Asymptotic per-epoch training cost for representative heads.

| Loss | Per-epoch cost | Notes |
|------|----------------|-------|
| Softmax | $\Theta(NC)$ | Dense EXP/LOG/DIV |
| Sparsemax | $\Theta(N\,C\log C)$ | Threshold via sorting |
| $\alpha$-Entmax | $\Theta(N\,C\log C + N|\mathcal{P}|)$ | Threshold sort + support-wise backprop |
| Rankmax | $\Theta(NC)$ | When setting to standard simplex |
| Sampled Softmax | $\Theta(Nk)$ | $(k+1)$ classes/update; sampling $O(1)$ |
| NCE | $\Theta(Nk)$ | One pos $+\ k$ neg logistic terms; sampling $O(1)$ |
| HSM | $\Theta(N\log C)$ | Depth $\simeq \lceil\log C\rceil$ with tree construction $O(|V|)$ |
| RG-ALS | — | Dominated by solving closed-form solutions |

**Q3.** Do the computational complexity analysis translate into observable differences in training time across varying model sizes and architectures?

While Softmax variants have been extensively studied in NLP tasks (Niculae et al., 2018; Peters et al., 2019; Correia et al., 2019), we follow the setups of Kong et al. (2020) and Pu et al. (2025) to focus on the domain of recommender systems. This domain is particularly suitable as it emphasizes both classification metrics (P@$k$, R@$k$) and ranking metrics (N@$k$), involves large-scale sparse data where gradient dynamics are more salient, and supports diverse backbones spanning matrix factorization, sequential, and graph-based models. Moreover, recommendation systems are of high practical relevance due to wide industrial adoption and tangible user impact.

We evaluate three public benchmarks (**ML-1M, Electronics, Gowalla**) with three representative backbones: **MF**, **SASRec**, and **LightGCN**. Unless otherwise stated, we apply identical training protocols across methods, including batch size, optimizer, and regularization. We compare both exact surrogates (**Softmax, Sparsemax, $\alpha$-Entmax, Rankmax**) and approximate methods (**SSM, NCE, HSM, RG**). Implementation details are provided in Appendix C.1.

**Q1: Accuracy under aligned training protocols.** For each (dataset, backbone), we evaluate all surrogates under identical hyperparameters: learning rates $\{10^{-3}, 5 \times 10^{-4}, 10^{-4}\}$ with model selection based on validation N@20, P@20, and R@20. Results are reported in Appendix C.2.

**Q2: Bias–variance decomposition vs. empirical behavior.** For sampling-based approximations, we sweep $k \in \{5, 10, 50, 100\}$ and proposal distributions $Q \in \{\text{Uniform}, \text{Dynamic Negative Sampling (DNS)}\}$, and compare against validation metrics in Appendix C.3.

**Q3: Training-time efficiency.** We measure both per-epoch and cumulative wall-clock time, and report *metric vs. epoch* and *metric vs. time* curves in Appendix C.4.

**Gradient visualization.** In addition, we visualize gradient matrices with heatmaps, which highlight dense competitive gradients for SOP and extended flat regions for sparse WOP losses in Appendix C.5.

# 5 RELATED WORKS

A detailed discussion of existing approaches has been provided in Section 2; here we distill the most relevant directions that directly connect to our study.

**Statistical foundations.** The Softmax loss has been the centerpiece of theoretical investigation for decades, with its statistical consistency in both classification and ranking rigorously established through a sequence of works (Zhang, 2004b; Bartlett et al., 2006; Cossock & Zhang, 2008; Calauzenes et al., 2012; Ravikumar et al., 2011; Lapin et al., 2016; Yang & Koyejo, 2020). Building on these results, the Fenchel–Young framework (Blondel et al., 2019) provided a principled unification that not only recovers Softmax as a special case but also motivates alternative constructions such as Sparsemax (Martins & Astudillo, 2016), $\alpha$-Entmax (Peters et al., 2019), and Rankmax (Kong et al., 2020). These sparse variants have been explored in applications ranging from structured pre-

diction to neural sequence modeling, offering advantages in controllable sparsity, interpretability, and alignment with human-centric evaluation (Niculae et al., 2018; Correia et al., 2019).

**Computational scalability.** In parallel, the prohibitive cost of computing the log-partition in large output spaces has driven an extensive line of research on approximations. Sampling-based strategies includeNCE (Gutmann & Hyvärinen, 2010) and SSM (Jean et al., 2015), which trade variance for efficiency. Deterministic approaches, by contrast, exploit structural decompositions such as HSM (Morin & Bengio, 2005; Mikolov et al., 2013b) or analytic surrogates based on Taylor expansions (Banerjee et al., 2020). More recent developments investigate kernel-based or adaptive feature maps to scale further in extreme classification regimes (Blanc & Rendle, 2018).

## 6 CONCLUSION

This work develops a unified theoretical framework of the Softmax-family losses. Through the analysis of Softmax, Sparsemax, $\alpha$-Entmax and Rankmax in the Fenchel–Young framework, we establish their consistency properties both in classification and ranking scenarios. Besides, we analyze their convergence behaviors through the properties of Jacobian matrix, which clarifies why Softmax enjoys smooth optimization while sparse variants often struggle from defective gradient dynamics.

For sampling-based approximations, we introduce a bias–variance decomposition that makes explicit the trade-off between computational costs and statistical fidelity. Complementary complexity analysis further highlights how different choices balance accuracy and efficiency in large-class learning.

Overall, our results bridge statistical guarantees with optimization dynamics and computational constraints, offering both theoretical clarity and practical guidelines for selecting surrogate losses in extreme classification and ranking tasks. Together, these findings yield the following practical takeaways:

- When the number of classes $C$ is moderate and exact methods are feasible: Softmax provides SOP and a smaller Jacobian spectral norm (higher smoothness), which corresponds to better-conditioned optimization and faster convergence compared with its sparse variants, and should therefore be tested as the default choice.

- When $C$ is extremely large and approximate methods are required: The choice among approximate softmax surrogates can be guided by the bias–variance decomposition. These results quantify how each approximation deviates from the exact Softmax risk, enabling the selection of surrogate with the smallest total deviation and indicating how bias can be further reduced, e.g., by increasing sample size $k$ or adapting the proposal distribution $Q$.

### ETHICS STATEMENT

This work is theoretical and does not involve human subjects or sensitive data. However, improvements to loss functions may be applied in downstream systems such as language models and recommendation, which could amplify biases or fairness concerns present in training data. We also note that large-scale training with softmax approximations has environmental impact, and encourage responsible and sustainable use of the proposed methods.

### REPRODUCIBILITY STATEMENT

The codes are available at `https://anonymous.4open.science/r/ICLR-C078`. All datasets used in this work (ML-1M, Amazon-Electronics, Gowalla) are publicly available, and we provide download links and preprocessing scripts to ensure consistency with our setup. Detailed hyperparameters (e.g., learning rate, batch size, optimizer, negative sample size $k$, and proposal distribution $Q$ and training settings are reported in the appendix. We report averaged results over multiple random seeds and include standard deviations to account for variance.

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

APPENDIX

## A  FOUNDATIONS

### A.1  EVALUATION METRICS

The ultimate objective of ML model is to excel at task-specific evaluation metrics. These metrics, usually non-convex and discontinuous, evaluates a model's performance and are the actual quantities we wish to optimize. They operate directly on the model's output scores $s$ to measure performance against the ground-truth $y$, yet their direct optimization is computationally intractable. This challenge motivates using a smooth surrogate losses for training.

As for classification task, where an input belongs to exactly one class, let $y^* = \underset{i \in \{1, \cdots, C\}}{\mathrm{argmax}}\ y_i$, the natural and fundamental 0-1 error is given as:

$$\ell_{\mathrm{Top}-1}(\boldsymbol{y}, \boldsymbol{s}) = \mathbf{1}\left\{\exists j \neq y^* \text{ s.t. } s_j \geq s_{y^*}\right\}. \tag{14}$$

A vast body of literature has been dedicated to analyzing whether minimizing a given surrogate is **consistent** with minimizing the 0-1 loss. Zhang (2004b) and Bartlett et al. (2006) established the property of **classification-calibration**, which demonstrates whether a classifier learned via the surrogate will converge to the Bayes-optimal classifier with respect to the 0-1 loss. Beyond this, practical applications with a large number of classes often employ more lenient criteria, motivating the shift to the Top-$k$ error, a more forgiving metric that equals 1 if the true class is not among the $k$ highest-scoring classes:

$$\ell_{\mathrm{Top}-k}(\boldsymbol{y}, \boldsymbol{s}) = \mathbf{1}\left\{y^* \notin \mathrm{Top}_k(\boldsymbol{s})\right\}, \tag{15}$$

where $\mathrm{Top}_k(\boldsymbol{s})$ is the set of indices of the top $k$ scores. Discussions of **Top-$k$ calibration** can be found in Lapin et al. (2016); Yang & Koyejo (2020). While the Top-$k$ error is intuitive for multiclass problems, recent theoretical advances (Menon et al., 2019) have revealed that surrogate losses optimizing Top-$k$ error serve as principled multi-label **reductions**, which can be transformed into a multi-label surrogate loss consistent with metrics like Precision@$k$ (**P@$k$**) or Recall@$k$(**R@$k$**). This insight provides a powerful bridge, reframing a classic classification metric as a key technique for tackling more complex multi-label scenarios. Let $\pi(\boldsymbol{s})$ be the permutation of indices $\{1, \ldots, C\}$ that sorts $\boldsymbol{s}$ in descending order, $|\boldsymbol{y}| = \sum_i y_i$ be the total number of relevant items,

$$\mathrm{P@}k(\boldsymbol{y}, \boldsymbol{s}) = \frac{1}{k}\sum_{i=1}^{k} y_{\pi(i)}, \qquad \mathrm{R@}k(\boldsymbol{y}, \boldsymbol{s}) = \frac{1}{|\boldsymbol{y}|}\sum_{i=1}^{k} y_{\pi(i)}. \tag{16}$$

As for the domain of ranking, which is critical for industrial applications like search and recommendation where the precise order of predictions is paramount. A standard metric for this setting is **DCG** (or its normalised version **NDCG**), a position-sensitive metric that rewards placing more relevant items at higher ranks. Its truncated version with cutoff at $k$ is given as:

$$\mathrm{N@}k(\boldsymbol{y}, \boldsymbol{s}) = \frac{\mathrm{DCG\,@}k(\boldsymbol{y}, \boldsymbol{s})}{\mathrm{IDCG\,@}k(\boldsymbol{y})}, \quad \text{where} \quad \mathrm{DCG\,@}k = \sum_{i=1}^{k} \frac{y_{\pi(i)}}{\log_2(i+1)}. \tag{17}$$

The corresponding loss, $\ell_{\mathrm{NDCG}} = 1 - \mathrm{NDCG}$, is a cornerstone of modern ranking systems. Choosing **DCG** as a main focus is a good choice from theoretical perspectives. Positive results have shown that for specific ranking surrogates, consistency with **DCG** is indeed achievable (Ravikumar et al., 2011; Cossock & Zhang, 2008). On the other hand, Calauzenes et al. (2012) proves that **NO** convex surrogate loss is calibrated for other familiar ranking metrics (including Average Precision and Mean Reciprocal Rank). Thus, **DCG** becomes the canonical and most representative metric for the rigorous analysis of ranking surrogates.

Since these evaluation metrics are discrete, they cannot be optimized directly with gradient-based methods. Instead, ML training process relies on minimizing a smooth, convex surrogate loss, in which the softmax cross-entropy is a good choice.

### A.2  FENCHEL–YOUNG LOSS FRAMEWORK

The Fenchel–Young (F–Y) framework(Blondel et al., 2019) provides a unifying recipe for constructing convex and classification-calibrated losses from a convex regularizer. Let $\Omega : \Delta^C \to \mathbb{R}$ be a

convex potential function defined over the probability simplex $\Delta^C$. Its Fenchel conjugate is

$$\Omega^*(\boldsymbol{s}) = \sup_{\boldsymbol{p} \in \Delta^C} \{\langle \boldsymbol{s}, \boldsymbol{p} \rangle - \Omega(\boldsymbol{p})\}, \quad \boldsymbol{s} \in \mathbb{R}^C \tag{18}$$

**Definition A.1** (Fenchel-Young Loss). Given a label $\boldsymbol{y} \in \{0,1\}^C$, a Fenchel–Young loss is defined as

$$\mathcal{L}_\Omega(\boldsymbol{y}, \boldsymbol{s}) = \Omega^*(\boldsymbol{s}) + \Omega(\boldsymbol{y}) - \langle \boldsymbol{s}, \boldsymbol{y} \rangle. \tag{19}$$

The predicted probabilities are obtained from the gradient of the conjugate,

$$\hat{\boldsymbol{p}}(\boldsymbol{s}) = \nabla \Omega^*(\boldsymbol{s}), \tag{20}$$

and the gradient of the loss has the simple form

$$\nabla_{\boldsymbol{s}} \ell_\Omega(\boldsymbol{y}, \boldsymbol{s}) = \hat{\boldsymbol{p}}(\boldsymbol{s}) - \boldsymbol{y}. \tag{21}$$

The F–Y construction guarantees convexity, smooth optimization, and classification-calibration (Fisher-consistency)(Williamson et al., 2016; Blondel et al., 2019). If $\Omega(\boldsymbol{y}) = \sum_i y_i \log y_i$ (negative Shannon entropy), then its conjugate is $\Omega^*(\boldsymbol{s}) = \log \sum_{i=1}^C \exp(s_i)$, and the predicted probability is the softmax function in Eq.(1). The F–Y loss recovers exactly Softmax loss.

The F-Y framework provides a principled and constructive alternative on analyzing and designing surrogate losses. Before its introduction, the design of loss functions in machine learning was largely heuristic, with activation functions and training losses often employed separately. F-Y losses, however, start from a single convex regularizer $\Omega$, which jointly derives both a prediction map $\hat{\boldsymbol{y}} = \nabla \Omega^*(\boldsymbol{s})$ and a convex, differentiable surrogate loss $\mathcal{L}_\Omega$. Once $\Omega$ is specified, the resulting loss follows automatically, and its gradient is guaranteed to take the residual form $\hat{y} - y$. This unifies the design of activation functions and losses under the same object, and enables choosing corresponding surrogates for complex or sparse output spaces.

**Other examples.** Alternative choices of $\Omega$ yield new mappings and loss families:

- If $\Omega(\boldsymbol{p}) = \frac{1}{2}\|\boldsymbol{p}\|_2^2$, one obtains *Sparsemax* (Martins & Astudillo, 2016), which projects scores $s$ onto the simplex via Euclidean projection, yielding sparse probability distributions.

- If $\Omega$ is a Tsallis $\alpha$-entropy (Tsallis, 1988), one obtains the $\alpha$-*Entmax* family (Peters et al., 2019), which interpolates between softmax ($\alpha = 1$) and sparsemax ($\alpha = 2$), producing distributions of controllable sparsity.

### A.3 BREGMAN DIVERGENCE

A powerful tool for analyzing consistency is the Bregman divergence (Bregman, 1967). Given a continuously-differentiable and strictly convex function $\Omega : \mathbb{R}^C \to \mathbb{R}$, the Bregman divergence $D_\Omega : \mathbb{R}^C \times \mathbb{R}^C \to \mathbb{R}$ is defined as:

$$D_\Omega(\boldsymbol{p}, \boldsymbol{q}) = \Omega(\boldsymbol{p}) - \Omega(\boldsymbol{q}) - \langle \nabla \Omega(\boldsymbol{q}), \boldsymbol{p} - \boldsymbol{q} \rangle. \tag{22}$$

The significance of this tool lies in its direct connection to Bayes-optimal predictors under a given loss. The goal of a learning algorithm is to find a function $f$ that minimizes the expected surrogate risk,

$$R_{\mathcal{L}}(f) = \mathbb{E}_{(\boldsymbol{x}, \boldsymbol{y}) \sim D}[\mathcal{L}(\boldsymbol{y}, f(\boldsymbol{x}))] = \mathbb{E}_{\boldsymbol{x}}\left[\mathbb{E}_{\boldsymbol{y} \sim P(\boldsymbol{y}|\boldsymbol{x})}[\mathcal{L}(\boldsymbol{y}, f(\boldsymbol{x}))]\right]. \tag{23}$$

To minimize the global risk, one can minimize the inner expectation for each point $\boldsymbol{x}$ independently. This defines the pointwise Bayes-optimal prediction $\boldsymbol{p}^*(\boldsymbol{x})$ for a model that outputs a probability vector:

$$\boldsymbol{p}^* = \arg\min_{\hat{\boldsymbol{p}} \in \Delta^C} \mathbb{E}_{\boldsymbol{y} \sim \boldsymbol{\eta}(\boldsymbol{x})}[\mathcal{L}(\boldsymbol{y}, \hat{\boldsymbol{p}})], \tag{24}$$

where $\boldsymbol{\eta}(\boldsymbol{x})$ is the true conditional probability vector $[\mathbb{P}(y = i|\boldsymbol{x})]_{i=1}^C$. Here we naturally replace the variable $\boldsymbol{s} = f(\boldsymbol{x})$ with $\hat{\boldsymbol{p}}$. If a surrogate loss can be expressed as a Bregman divergence, this minimization has a unique solution (Reid & Williamson, 2011; Blondel et al., 2019). Given that $D_\Omega(\boldsymbol{p}, \boldsymbol{q}) \geq 0$ with equality holding if and only if $\boldsymbol{p} = \boldsymbol{q}$. Therefore, the unique minimizer is:

$$\boldsymbol{p}^* = \boldsymbol{\eta}(\boldsymbol{x}). \tag{25}$$

This direct alignment—showing that minimizing the risk forces the prediction to match the true posterior—is the cornerstone of proving consistency for various ML tasks.

## A.4 CONNECTION TO MIRROR DESCENT

Fenchel–Young losses admit a natural interpretation under the geometry of Mirror Descent. Given a convex potential $\Omega$, MD updates follow

$$\boldsymbol{x}_{t+1} = \arg\min_{\boldsymbol{x}} \langle \nabla f(\boldsymbol{x}_t), \boldsymbol{x} \rangle + \frac{1}{\eta} D_\Omega(\boldsymbol{x}, \boldsymbol{x}_t), \tag{26}$$

where $D_\Omega$ is the Bregman divergence induced by $\Omega$. MD performs gradient steps in the dual geometry defined by $\Omega$, with the mirror map $\nabla\Omega$ governing how gradients are transported between these spaces. The optimality condition of MD implies

$$\boldsymbol{s}^* = \nabla\Omega(\boldsymbol{x}^*) \Leftrightarrow \nabla\Omega^*(\boldsymbol{s}^*) = \boldsymbol{x}^*. \tag{27}$$

Similarly, a Fenchel–Young loss generated by $\Omega$ enforces this same condition in supervised learning:

$$\nabla_{\boldsymbol{s}} L_\Omega(\boldsymbol{y}, \boldsymbol{s}) = \nabla\Omega^*(\boldsymbol{s}) - \boldsymbol{y}, \tag{28}$$

driving the model toward $\hat{\boldsymbol{y}} = \nabla\Omega^*(\boldsymbol{s}) = \boldsymbol{y}$. Thus, F–Y losses can be viewed as supervised analogues of MD's mirror-space optimality under same geometry. The choice of $\Omega$ simultaneously connects the model's geometry, its prediction rule, and the structure of its induced supervised loss.

## A.5 SOFTMAX APPROXIMATIONS

**Sampling-based Methods.**

**Sampled Softmax.** Jean et al. (2015) proposed an efficient approximation by restricting normalization to the true class $y$ and a set of $M$ sampled negatives $\{y_1, \ldots, y_M\}$:

$$\mathcal{L}_{\text{SSM-Simple}}(\boldsymbol{y}, \boldsymbol{s}) = -\sum_{i:y_i=1} \left( \log \frac{\exp(s_{y_i})}{\exp(s_{y_i}) + \sum_{j=1}^{M} \exp(s_{y_j})} \right). \tag{29}$$

This formulation reduces computational complexity from $O(C)$ to $O(M)$, but introduces a bias since the partition function is no longer computed over all classes. To handle with, a bias correction is often applied. Specifically, if the negatives are drawn from a proposal distribution $q(\cdot)$, the logits of the sampled classes are adjusted by subtracting $\log q(y_j)$:

$$\tilde{s}_{y_j} = s_{y_j} - \log q(y_j), \quad j = 1, \ldots, M, \tag{30}$$

and the corrected sampled softmax loss becomes

$$\mathcal{L}_{\text{SSM}}(\boldsymbol{y}, \boldsymbol{s}) = -\sum_{i:y_i=1} \left( \log \frac{\exp(\tilde{s}_{y_i})}{\exp(\tilde{s}_{y_i}) + \sum_{j=1}^{M} \exp(\tilde{s}_{y_j})} \right) \tag{31}$$

which ensures the gradient an unbiased estimator of the full softmax gradient in expectation over the sampling distribution $q$.

**NCE.** Gutmann & Hyvärinen (2010) reformulates density estimation as a binary classification problem distinguishing true samples $(s_{y_i}, y_i)$ from noise samples $(s_{y'}, y')$ drawn from a fixed distribution $q(\cdot)$. The surrogate loss is

$$\mathcal{L}_{\text{NCE}}(\boldsymbol{y}, \boldsymbol{s}) = -\sum_{i:y_i=1} \left( \log \sigma\big(s_{y_i} - \log(kq(y_i))\big) - \sum_{y'\sim q} \log \sigma\big(-s_{y'} + \log(kq(y'))\big) \right), \quad (32)$$

where $\sigma(\cdot)$ is the sigmoid and $k$ is the number of negatives. It's worthy noting that NCE is asymptotically consistent with maximum likelihood estimation.

**Deterministic Approximations.**

**(I) Hierarchical Softmax.** Morin & Bengio (2005) replaces the Softmax mapping over global scores $\{s_j\}_{j=1}^{C}$ with a tree factorization. Let $\text{path}(y_i) = (u_1, \ldots, u_{L_i})$ be the path to label $y_i$, and

$\mathcal{C}(u)$ the children of node $u$. At each internal node $u$, HSM uses node-local scores $s_u(j)$, $j \in \mathcal{C}(u)$, to form a local softmax

$$p(j \mid u) = \frac{\exp\big(s_u(j)\big)}{\sum_{k \in \mathcal{C}(u)} \exp\big(s_u(k)\big)}, \tag{33}$$

and the class probability is the product along the path

$$\hat{\boldsymbol{p}}_{\mathrm{HSM}}(y_i) = \prod_{u \in \mathrm{path}(y)} p(c(u) \mid u). \tag{34}$$

The training objective is the (exact) negative log-likelihood

$$\mathcal{L}_{\mathrm{HSM}}(\boldsymbol{y}) = -\sum_i \log \hat{\boldsymbol{p}}_{\mathrm{HSM}}(y_i). \tag{35}$$

Crucially, unlike other Softmax-family losses, which map global score vectors ($s_j = \mathrm{sim}(\boldsymbol{u}, \boldsymbol{v}_j)$) to a distribution, HSM uses node-specific logits $s_u(\cdot)$; it is therefore a different output parameterization rather than a drop-in loss on the same $\{s_j\}$, which cannot be naturally plugged into different backbones.

**(II) Taylor Approximations.** Banerjee et al. (2020) proposed *Taylor-softmax*, which approximates the exponential in softmax by a finite-order Taylor expansion. Specifically, $e^{s_i}$ is replaced with a low-order polynomial $1 + s_i + \frac{1}{2} s_i^2$, followed by normalization across classes. This yields a probability distribution that is computationally cheaper and empirically competitive with the standard softmax on classification benchmarks. Higher-order expansions and variants such as margin-based Taylor-softmax were also considered, showing that truncated polynomial approximations can serve as viable surrogates to the exponential mapping. More recently, Pu et al. (2025) applied a Taylor expansion directly to Softmax loss. Expanding the log-partition function $\log \sum_j e^{s_j}$ around the origin yields a closed-form quadratic surrogate:

$$\hat{Z}_{\mathrm{RG}}(\boldsymbol{s}) = \log C + \frac{1}{C} \sum_{j=1}^{C} s_j + \frac{1}{2} \boldsymbol{s}^\top \left( \frac{1}{C} I - \frac{1}{C^2} \mathbf{1} \mathbf{1}^\top \right) \boldsymbol{s}. \tag{36}$$

The surrogate loss is

$$\mathcal{L}_{\mathrm{RG}}(\boldsymbol{y}, \boldsymbol{s}) = -\sum_{i:y_i=1} \left( s_i - \hat{Z}_{\mathrm{RG}}(\boldsymbol{s}) \right). \tag{37}$$

This loss eliminates the need to compute the partition function $Z(\boldsymbol{s})$ and can be optimized efficiently via alternating least squares (ALS).

## B    SUPPLEMENTARIES OF MAIN RESULTS

### B.1    PROOF OF PROPOSITION 3.3

**Proposition B.1.** $\forall k > 1$, *Sparsemax and $\alpha$-Entmax are Top-$k$ calibrated and DCG-consistent.*

*Proof.* We first present the proof for the Sparsemax case and then generalize. Let $\boldsymbol{s} \in \mathbb{R}^C$ be a vector of scores and $\boldsymbol{p} \in \Delta^C$ be the true conditional probability distribution. Let $\boldsymbol{y} \in \{0,1\}^C$ be a one-hot label vector drawn according to the distribution $\boldsymbol{p}$.

**1. Bayes Optimality Condition**    The Sparsemax loss, $\mathcal{L}_{\mathrm{sparsemax}}(\boldsymbol{y}, \boldsymbol{s})$, is derived from the regularizer $\Omega(\boldsymbol{p}) = \frac{1}{2} \|\boldsymbol{p}\|_2^2$. The predicted probability vector is given by the gradient of the conjugate, $\hat{\boldsymbol{p}}(\boldsymbol{s}) = \nabla \Omega^*(\boldsymbol{s}) = \mathrm{sparsemax}(\boldsymbol{s})$.

The pointwise conditional risk is the expected loss $\mathcal{L}(\boldsymbol{s}; \boldsymbol{p}) = \mathbb{E}_{\boldsymbol{y} \sim \boldsymbol{p}}[\mathcal{L}_{\mathrm{smax}}(\boldsymbol{y}, \boldsymbol{s})]$. From the F–Y gradient identity, we have:

$$\nabla_{\boldsymbol{s}} \mathcal{L}(\boldsymbol{p}; \boldsymbol{s}) = \hat{\boldsymbol{p}}(\boldsymbol{s}) - \boldsymbol{p} = \mathrm{sparsemax}(\boldsymbol{s}) - \boldsymbol{p}. \tag{38}$$

Setting the gradient to zero, any Bayes-optimal score vector $\boldsymbol{s}^*$ that minimizes the risk must satisfy the optimality condition:

$$\mathrm{sparsemax}(\boldsymbol{s}^*) = \boldsymbol{p}. \tag{39}$$

**2. Structure of the Bayes-Optimal Solution**    Let $\mathcal{P} := \{i \mid p_i > 0\}$ be the support of the true distribution $\boldsymbol{p}$. The optimality condition in Eq.(39) imposes a clear structure on the optimal scores $\boldsymbol{s}^*$. By the definition of Sparsemax, there must exist a threshold $\tau^*$ such that:

$$\forall i \in \mathcal{P}, \quad s_i^* - \tau^* = p_i \implies s_i^* = p_i + \tau^* \tag{40}$$

$$\forall j \notin \mathcal{P}, \quad s_j^* - \tau^* \leq 0 \implies s_j^* \leq \tau^* \tag{41}$$

From this structure, two critical properties emerge:

  *(i)* **Support Separation:** The scores of classes in the true support are strictly greater than the scores of classes outside the support.

$$\min_{i \in \mathcal{P}} s_i^* = (\min_{i \in \mathcal{P}} p_i) + \tau^* > \tau^* \geq \max_{j \notin \mathcal{P}} s_j^*. \tag{42}$$

  *(ii)* **Order Preservation within Support:** Within the set of support classes, the scores are perfectly rank-ordered according to their true probabilities.

$$\forall i, j \in \mathcal{P}, \quad s_i^* - s_j^* = (p_i + \tau^*) - (p_j + \tau^*) = p_i - p_j. \tag{43}$$

  This directly implies $s_i^* > s_j^* \iff p_i > p_j$.

**3. Top-$k$ Calibration**    Recall that $\mathrm{Top}_k(\boldsymbol{v})$ denote the set of indices of the top $k$ components of any vector $\boldsymbol{v}$. Given the Bayes-optimal decision $\mathrm{Top}_k(\boldsymbol{p})$, we show that $\mathrm{Top}_k(\boldsymbol{s}^*)$ is a Bayes-optimal set for any $k$.

  - **Case $k \leq |\mathcal{P}|$:** The top $k$ classes of $\boldsymbol{p}$ are all within the support $\mathcal{P}$. By property *(ii)*, the relative order of scores $s_i^*$ for all $i \in \mathcal{P}$ exactly matches the order of probabilities $p_i$. Thus, $\mathrm{Top}_k(\boldsymbol{s}^*) = \mathrm{Top}_k(\boldsymbol{p})$.

  - **Case $k > |\mathcal{P}|$:** By property *(i)*, all scores in $\mathcal{P}$ are ranked strictly higher than all scores not in $\mathcal{P}$. Therefore, $\mathrm{Top}_k(\boldsymbol{s}^*)$ must contain all of $\mathcal{P}$. The remaining $k - |\mathcal{P}|$ positions are filled by indices from outside $\mathcal{P}$. Since all $j \notin \mathcal{P}$ have $p_j = 0$, they are all tied for the lowest rank, and any selection constitutes a Bayes-optimal completion.

In both cases, $\mathrm{Top}_k(\boldsymbol{s}^*)$ is a Bayes-optimal set, proving Top-$k$ calibration.

**4. DCG-Consistency**    The expected DCG is maximized by ranking classes in non-increasing order of their true probabilities $p_i$. The optimal scores $\boldsymbol{s}^*$ induce exactly such a ranking. Property *(ii)* ensures the correct ordering within the support $\mathcal{P}$, and property *(i)* ensures that all classes in $\mathcal{P}$ are ranked strictly before all classes not in $\mathcal{P}$. Therefore, the ranking from $\boldsymbol{s}^*$ is Bayes-optimal for DCG.

**Generalization to $\alpha$-Entmax**    The proof for the F–Y loss derived from the $\alpha$-Entmax regularizer, $\Omega_\alpha(\boldsymbol{p})$, follows the same structure. The optimality condition becomes $\mathrm{entmax}_\alpha(\boldsymbol{s}^*) = \boldsymbol{p}$. The structure of the solution for $i \in \mathcal{P}$ also yields the same **Support Separation** and **Order Preservation within Support** properties. As both crucial properties are preserved, the conclusions for Top-$k$ calibration and DCG-consistency follow directly.    $\square$

### B.2    THEORETICAL UNDERSTANDINGS OF WOP

We provide formal results quantifying the effect of WOP losses on ranking-based metrics and optimization dynamics. We first establish a lower bound on expected DCG degradation due to ties, then analyze the gradient bias induced by sparse WOP mappings.

#### B.2.1    EXPECTED DCG LOWER BOUND UNDER TIES

**Theorem B.2** (Tie-induced expected DCG loss lower bound). *Fix a block of tied scores occupying positions $\{z+1, \ldots, z+m\}$ in the ranking induced by s, with $m \in \mathbb{N}$. Let $r := \sum_{t=1}^m y_{\pi(z+t)}$ be*

*the number of relevant items in this block. Let $w_i := 1/\log_2(i+1)$. Then, relative to the block-wise optimal arrangement (placing the $r$ relevant items at the earliest $r$ positions of this block), any tie-breaking scheme whose expectation is uniform over the $m!$ permutations satisfies*

$$\mathbb{E}[\mathrm{DCG}_{\mathrm{block}}] = \frac{r}{m} \sum_{i=1}^{m} w_{z+i}, \qquad \mathrm{DCG}_{\mathrm{block}}^{\star} = \sum_{i=1}^{r} w_{z+i}. \tag{44}$$

*Therefore the expected DCG loss obeys*

$$\Delta\,\mathrm{DCG}_{\mathrm{block}} := \mathrm{DCG}_{\mathrm{block}}^{\star} - \mathbb{E}[\mathrm{DCG}_{\mathrm{block}}] \geq \sum_{i=1}^{r} w_{z+i} - \frac{r}{m} \sum_{i=1}^{m} w_{z+i} \geq 0. \tag{45}$$

*For normalized NDCG, dividing both sides by $\mathrm{IDCG}(y)$ gives the corresponding lower bound.*

*Proof.* Inside the block, each position is occupied by a relevant item with probability $r/m$. By linearity of expectation, the expected DCG contribution is $\frac{r}{m} \sum_{i=1}^{m} w_{z+i}$. The block-optimal DCG is obtained by placing all $r$ relevant items at the earliest slots, giving $\sum_{i=1}^{r} w_{z+i}$. Subtracting yields the bound, and monotonicity of $w_i$ ensures nonnegativity. $\qquad\square$

### B.2.2 Gradient bias of sparse WOP mappings

We measure alignment with the improvement direction of a DCG-consistent surrogate, defined as

$$d_{\mathrm{DCG}}(\boldsymbol{y}, \boldsymbol{s}) := -\nabla\mathcal{L}(\boldsymbol{y}, \boldsymbol{s}), \tag{46}$$

which ensures that larger inner products indicate better alignment with DCG-improving updates.

**Proposition B.3** (Sparse WOP: reduced alignment with DCG improvement direction). *Let $\hat{p}(\boldsymbol{s})$ be the prediction of a WOP sparse Fenchel–Young mapping (e.g., Sparsemax/Entmax), with loss gradient*

$$\mathrm{grad}_{\mathrm{FY}}(\boldsymbol{y}, \boldsymbol{s}) = \hat{p}(\boldsymbol{s}) - \boldsymbol{y}. \tag{47}$$

*Let $\pi$ sort $\boldsymbol{s}$ in descending order. Consider a DCG-consistent surrogate admitting a pairwise gradient form*

$$d_{\mathrm{DCG}}(\boldsymbol{s}; \boldsymbol{y}) = \sum_{i:\, y_{\pi(i)}=1} \sum_{j:\, y_{\pi(j)}=0} \alpha_{i,j} \left( \boldsymbol{e}_{\pi(j)} - \boldsymbol{e}_{\pi(i)} \right), \qquad \alpha_{i,j} \geq c \cdot w_{\pi(i)} \;\; (c > 0), \tag{48}$$

*with position weights $w_r = 1/\log_2(r+1)$. Define the index set of zeroed negative components*

$$\mathcal{Z}(\boldsymbol{s}) := \{\, j :\; \hat{p}_j(\boldsymbol{s}) = 0, y_j = 0 \,\}. \tag{49}$$

*Construct a comparator gradient $\mathrm{grad}_{\mathrm{FY}}^{(+)}$ that coincides with $\mathrm{grad}_{\mathrm{FY}}$, except that for $j \in \mathcal{Z}(\boldsymbol{s})$, we replace zero entries by nonnegative values $\tilde{p}_j(\boldsymbol{s}) \geq 0$. Then*

$$\left\langle \mathrm{grad}_{\mathrm{FY}}(\boldsymbol{y}, \boldsymbol{s}),\, d_{\mathrm{DCG}}(\boldsymbol{y}, \boldsymbol{s}) \right\rangle \leq \left\langle \mathrm{grad}_{\mathrm{FY}}^{(+)}(\boldsymbol{y}, \boldsymbol{s}),\, d_{\mathrm{DCG}}(\boldsymbol{y}, \boldsymbol{s}) \right\rangle - \sum_{j \in \mathcal{Z}(\boldsymbol{s})} \sum_{i:\, y_{\pi(i)}=1} \alpha_{i,j}\, \tilde{p}_j(\boldsymbol{s}), \tag{50}$$

*Proof.* For $j \in \mathcal{Z}(\boldsymbol{s})$, $(\mathrm{grad}_{\mathrm{FY}})_j = 0$ while $(\mathrm{grad}_{\mathrm{FY}}^{(+)})_j = \tilde{p}_j(\boldsymbol{s}) \geq 0$.

For each pair $(i, j)$ with $y_{\pi(i)} = 1$, $y_{\pi(j)} = 0$,

$$\left\langle \mathrm{grad}_{\mathrm{FY}}^{(+)} - \mathrm{grad}_{\mathrm{FY}},\, -\alpha_{i,j}(\boldsymbol{e}_{\pi(i)} - \boldsymbol{e}_{\pi(j)}) \right\rangle = \alpha_{i,j}\,(\mathrm{grad}_{\mathrm{FY}}^{(+)} - \mathrm{grad}_{\mathrm{FY}})_j = \alpha_{i,j}\,\tilde{p}_j(\boldsymbol{s}) \geq 0, \tag{51}$$

because the $\pi(i)$-coordinate cancels and only the $j$-coordinate contributes. Summing over all $(i, j)$ yields

$$\langle \mathrm{grad}_{\mathrm{FY}}^{(+)} - \mathrm{grad}_{\mathrm{FY}},\, d_{\mathrm{DCG}} \rangle = \sum_{j \in \mathcal{Z}(\boldsymbol{s})} \sum_{i:\, y_{\pi(i)}=1} \alpha_{i,j}\, \tilde{p}_j(\boldsymbol{s}) \geq 0, \tag{52}$$

$\qquad\square$

### B.3 WOP Sparse Alternative: Rankmax

Beyond Sparsemax and $\alpha$-Entmax, another sparse variant of F-Y losses is **Rankmax** proposed by Kong et al. (2020). Unlike Sparsemax/Entmax, which truncate all classes with full-score distribution, Rankmax explicitly leverages the score of the ground-truth class to determine the support. For the simplest version, Rankmax admits the following closed form:

$$\hat{p}_i^{\mathrm{rm}}(\boldsymbol{s}; y) \;=\; \frac{(s_i - s_y + 1)_+}{\sum_{j=1}^{C}(s_j - s_y + 1)_+}, \tag{53}$$

Thus Rankmax always assigns positive mass to the true class $y$, while aggressively zeroing out all classes whose scores lie below $s_y - 1$.

**Proposition B.4.** *Rankmax is WOP.*

*Proof.* If $s_i > s_j$, then $\hat{p}_i^{\mathrm{rm}} \geq \hat{p}_j^{\mathrm{rm}}$ always translates, whose equality only satisfies when $s_y + 1 > s_i > s_j$. Hence Rankmax does not satisfy SOP but WOP.

$\square$

Compared to Sparsemax and Entmax, Rankmax exhibits stronger reliance on the ground-truth: the normalization is explicitly centered at the true-class score $s_y$, making the gradient magnitude directly sensitive to the confidence in the correct label and ensuring that no hard negatives with $s_j \geq s_y$ are missed. In contrast, Sparsemax and $\alpha$-Entmax determine their threshold $\tau$ from the global distribution of logits, without focusing on $s_y$, and thus cannot reliably distinguish whether the true class is already highly confident or severely under-confident.

Besides, Softmax produces dense gradients, spreading the update budget over easy negatives and never truly halting even on large-margin examples. This may dilute the critical updates between the ground truth and hard negatives. Rankmax, although being sparse and WOP, concentrates updates on hard negatives while automatically stopping on high-confidence cases, thereby focusing more on critical comparisons for ranking tasks.

### B.4 Convergence Analysis

We now analyze the convergence of gradient descent by examining the curvature of the prediction mapping $\hat{\boldsymbol{p}}$. Since the Hessian satisfies

$$\nabla_s^2 \mathcal{L}(\boldsymbol{y}, \boldsymbol{s}) \;=\; \nabla_{\boldsymbol{s}}\hat{\boldsymbol{p}}(\boldsymbol{s}) \;=:\; J(\boldsymbol{s}), \tag{54}$$

the smoothness of the objective at the logit level is governed by the spectral norm $\|J(\boldsymbol{s})\|_2$. For a parametric model $\boldsymbol{s} = \boldsymbol{s}_\theta(x)$, the chain rule yields that the Hessian with respect to $\theta$ involves both $J(\boldsymbol{s})$ and the Jacobian $\partial \boldsymbol{s}/\partial \theta$. Consequently, an upper bound on the smoothness parameter of the loss is given by

$$L_\theta \;\leq\; \sup_{\boldsymbol{s}} \|J(\boldsymbol{s})\|_2 \cdot \sup_{x,\theta}\left\|\tfrac{\partial \boldsymbol{s}}{\partial \theta}\right\|_2^2. \tag{55}$$

In practice, it is common to add $\ell_2$-regularization $\frac{\gamma}{2}\|\theta\|_2^2$ with $\gamma > 0$, both to improve generalization and to stabilize optimization. From an optimization perspective, this modification adds $\gamma I$ to the parameter Hessian, ensuring that the overall objective is $\mu_\theta$-strongly convex with at least $\mu_\theta \geq \gamma$. At the same time, the smoothness parameter is shifted to

$$L_\theta \;\leq\; \sup_{\boldsymbol{s}} \|J(\boldsymbol{s})\|_2 \cdot \sup_{\theta}\left\|\tfrac{\partial \boldsymbol{s}}{\partial \theta}\right\|_2^2 \;+\; \gamma. \tag{56}$$

i.e. it belongs to the standard class of $(\mu_\theta, L_\theta)$-smooth strongly convex functions. For $L$-smooth function $f$, gradient descent $\theta_{t+1} = \theta_t - \eta \nabla f(\theta_t)$ satisfies $f(\theta_{t+1}) \leq f(\theta_t)$ for any step size $0 < \eta < 2/L$. If, in addition, $f$ is $\mu$-strongly convex, then choosing the constant step $\eta^\star = \frac{2}{L+\mu}$ yields the tight contraction(Nesterov, 1998),

$$\|\theta_t - \theta^\star\|_2 \;\leq\; \left(\tfrac{\kappa-1}{\kappa+1}\right)^t \|\theta_0 - \theta^\star\|_2, \quad \text{where } \kappa = \tfrac{L}{\mu} \tag{57}$$

The condition number of the regularized problem is then

$$\kappa := \frac{L_\theta}{\mu_\theta} \;\leq\; 1 + \frac{\sup_{\boldsymbol{s}} \|J(\boldsymbol{s})\|_2 \cdot \sup_\theta \|\partial \boldsymbol{s}/\partial\theta\|_2^2}{\gamma}. \tag{58}$$

**Lemma B.5** (Jacobian of Softmax). *For $\hat{p}_i(\boldsymbol{s}) = \exp(s_i)/\sum_j \exp(s_j)$ one has*

$$J_{\mathrm{sm}}(\boldsymbol{s}) \;=\; \mathrm{diag}(\hat{\boldsymbol{p}}) - \hat{\boldsymbol{p}}\hat{\boldsymbol{p}}^\top, \qquad \|J_{\mathrm{sm}}(\boldsymbol{s})\|_2 \;\leq\; \tfrac{1}{2}, \;\; \forall s.$$

*Proof.* For any unit vector $\boldsymbol{x} \in \mathbb{R}^C$, the Rayleigh quotient of $J_{\mathrm{sm}}$ coincides with the variance of some *r.v.* $X$

$$\boldsymbol{x}^\top J_{\mathrm{sm}}\boldsymbol{x} = \sum_i p_i x_i^2 - \Big(\sum_i p_i x_i\Big)^2 = \mathrm{Var}_p(X), \tag{59}$$

where $X$ taking value $x_i$ with probability $p_i$.

By Popoviciu's inequality,
$$\mathrm{Var}_p(X) \;\leq\; \tfrac{1}{4}(M-m)^2, \tag{60}$$

with $M = \max_i x_i$ and $m = \min_i x_i$. Since $\|\boldsymbol{x}\|_2 = 1$ implies $(M-m)^2 \leq (\sqrt{2}/2 + \sqrt{2}/2)^2 = 2$, we obtain

$$\boldsymbol{x}^\top J_{\mathrm{sm}}\boldsymbol{x} \;\leq\; \tfrac{1}{2}. \tag{61}$$

$$\square$$

As for Sparsemax, $\hat{p}_i(\boldsymbol{s}) = \max\{s_i - \tau(\boldsymbol{s}), 0\}$, we have

**Lemma B.6** (Jacobian of Sparsemax). *Within a fixed support region $\mathcal{P}$,*

$$J_{\mathrm{sp}}(\boldsymbol{s})\big|_{\mathcal{P}\times\mathcal{P}} = I_{|\mathcal{P}|} - \tfrac{1}{|\mathcal{P}|}\mathbf{1}\mathbf{1}^\top, \qquad J_{\mathrm{sp}}(\boldsymbol{s})\big|_{\mathcal{P}^c\times\mathbb{R}^C} = 0,$$

*whose spectrum is $\{1 \;(\text{with multiplicity } |\mathcal{P}| - 1),\; 0\}$. Thus*

$$\sup_{\boldsymbol{s}} \|J_{\mathrm{sp}}(\boldsymbol{s})\|_2 = 1. \tag{62}$$

*Proof.* The eigen-structure follows since $I_{|\mathcal{P}|} - \tfrac{1}{|\mathcal{P}|}\mathbf{1}\mathbf{1}^\top$ is the orthogonal projector onto $\mathbf{1}^\perp$. $\quad\square$

$\alpha$-Entmax admits the form $\hat{p}_i(\boldsymbol{s}) = \big((\alpha - 1)(s_i - \tau(\boldsymbol{s}))_+\big)^{\frac{1}{\alpha-1}}, \sum_i \hat{p}_i(\boldsymbol{s}) = 1$. Let $\boldsymbol{a} = [a_1, \cdots, a_{|\mathcal{P}|}]$ where $a_i := \hat{p}_i^{2-\alpha}$ on the support $\mathcal{P}$ and $S_{\boldsymbol{a}} := \sum_{i\in\mathcal{P}} a_i$.

**Lemma B.7** (Jacobian of $\alpha$-Entmax). *Within a fixed support region $\mathcal{P}$,*

$$J_\alpha(\boldsymbol{s})\big|_{\mathcal{P}\times\mathcal{P}} = \mathrm{diag}(\boldsymbol{a}) - \frac{\boldsymbol{a}\boldsymbol{a}^\top}{S_{\boldsymbol{a}}}, \qquad J_\alpha(\boldsymbol{s})\big|_{\mathcal{P}^c\times\mathbb{R}^C} = 0, \qquad a_i = \hat{p}_i^{2-\alpha}. \tag{63}$$

*Proof.* Define $u_i = (\alpha - 1)(s_i - \tau)$, so that $\hat{p}_i = \phi(u_i)$ with activation $\phi(u) = u_+^{1/(\alpha-1)}$. By the chain rule,

$$\frac{\partial \hat{p}_i}{\partial s_j} = \phi'(u_i)(\alpha - 1)(\delta_{ij} - \partial\tau/\partial s_j). \tag{64}$$

Since

$$\phi'(u) = \tfrac{1}{\alpha-1}\, u_+^{\frac{2-\alpha}{\alpha-1}} = \tfrac{1}{\alpha-1}\, \hat{p}_i^{2-\alpha}, \tag{65}$$

the factors of $(\alpha - 1)$ cancel, giving

$$\frac{\partial \hat{p}_i}{\partial s_j} = a_i\big(\delta_{ij} - \partial\tau/\partial s_j\big), \qquad a_i = \hat{p}_i^{2-\alpha}. \tag{66}$$

Differentiating the normalization constraint $\sum_{i\in\mathcal{P}} \hat{p}_i = 1$ yields

$$0 = \sum_{i\in\mathcal{P}} a_i(\delta_{ij} - \partial\tau/\partial s_j), \tag{67}$$

which implies $\partial \tau / \partial s_j = a_j / S_{\boldsymbol{a}}$ with $S_{\boldsymbol{a}} = \sum_{i \in \mathcal{P}} a_i$. Substituting back, we obtain the claimed block structure

$$J_\alpha(\boldsymbol{s})\big|_{\mathcal{P} \times \mathcal{P}} = \text{diag}(\boldsymbol{a}) - \frac{\boldsymbol{a}\boldsymbol{a}^\top}{S_{\boldsymbol{a}}}, \tag{68}$$

and the Jacobian vanishes outside the active support $\mathcal{P}$, completing the proof. $\square$

**Lemma B.8** (Lipschitz constant of $\alpha$-Entmax). *For $1 < \alpha \leq 2$, the Jacobian $J_\alpha(\boldsymbol{s})$ satisfies*

$$\sup_s \|J_\alpha(\boldsymbol{s})\|_2 \leq 1. \tag{69}$$

*Proof.* Within a fixed support $\mathcal{P}$, for any unit vector $\boldsymbol{x} \in \mathbb{R}^C$,

$$\boldsymbol{x}^\top J_\alpha \boldsymbol{x} = \sum_{i \in \mathcal{P}} a_i (x_i - \bar{x})^2, \qquad \bar{x} = \frac{\sum_{i \in \mathcal{P}} a_i x_i}{S_{\boldsymbol{a}}}. \tag{70}$$

Expanding the square and using $\sum_i a_i x_i = S_{\boldsymbol{a}} \bar{x}$ gives

$$\sum_i a_i (x_i - \bar{x})^2 = \sum_i a_i x_i^2 - S_{\boldsymbol{a}} \bar{x}^2 \leq \sum_i a_i x_i^2 \leq \Big(\max_{i \in \mathcal{P}} a_i\Big) \sum_i x_i^2 = \max_{i \in \mathcal{P}} a_i. \tag{71}$$

Hence $\boldsymbol{x}^\top J_\alpha \boldsymbol{x} \leq \max_i a_i$ for $\forall \boldsymbol{x}$, and therefore

$$\|J_\alpha(\boldsymbol{s})\|_2 = \sup_{\|\boldsymbol{x}\|_2 = 1} \boldsymbol{x}^\top J_\alpha \boldsymbol{x} \leq \max_{i \in \mathcal{P}} a_i. \tag{72}$$

Since $a_i = p_i^{2-\alpha} \in [0,1]$ for $1 < \alpha \leq 2$ and $p_i \in [0,1]$, we conclude $\|J_\alpha(\boldsymbol{s})\|_2 \leq 1, \forall s$, which proves the claim. $\square$

**Lemma B.9** (Jacobian of Rankmax). *On any region with fixed $\mathcal{P}$ (necessarily $y \in \mathcal{P}$),*

$$J_{\text{rm}}(s)\big|_{\mathcal{P} \times \mathcal{P}} = I_m - \frac{1}{m}\boldsymbol{1}\boldsymbol{1}^\top, \qquad J_{\text{rm}}(s)\big|_{\mathcal{P}^c \times [C]} = 0, \tag{73}$$

*hence*

$$\sup_s \|J_{\text{rm}}(\boldsymbol{s})\|_2 = 1. \tag{74}$$

Combining all lemmas with Eq.(58) gives, for any backbone with $L_G := \sup_{x,\theta} \|G(x,\theta)\|_2^2$,

**Softmax:** $\qquad \kappa \leq 1 + \frac{\frac{1}{2}L_G}{\gamma}, \qquad \eta_{\max} = \frac{2}{\frac{1}{2}L_G + \gamma}, \quad \eta^\star = \frac{2}{\frac{1}{2}L_G + 2\gamma}.$

**Sparse methods:** $\quad \kappa \leq 1 + \frac{L_G}{\gamma}, \qquad \eta_{\max} = \frac{2}{L_G + \gamma}, \quad \eta^\star = \frac{2}{L_G + 2\gamma}.$

Therefore, since GD's linear rate factor $\rho = \frac{\kappa - 1}{\kappa + 1}$ is monotonically increasing in $\kappa$, for the same backbone $L_G$ and regularization strength $\gamma$,

$$\rho_{\text{SM}} < \rho_{\alpha-\text{Entmax}} = \rho_{\text{Sparsemax}} = \rho_{\text{Rankmax}}, \tag{75}$$

Consequently, the parameter-level smoothness $L_\theta$ is uniformly smaller for softmax, yielding a better condition number $\kappa = L_\theta / \mu_\theta$ and thus a faster linear convergence rate of gradient descent. Beyond these exact F-Y losses, we also summarize softmax approximations, where the Jacobian structure and spectral norm bounds similarly determine their optimization behavior:

- **NCE:** $J_{\text{nce}}(s)$ is block-diagonal with logistic curvature terms; $\sup_s \|J_{\text{nce}}(\boldsymbol{s})\|_2 \leq \frac{1}{4}$.
- **Sampled Softmax(-Simple):** $J_{\text{ssm}}(s)$ coincides with the Fisher form of a local softmax on the sampled subset; $\sup_s \|J_{\text{ssm}}(\boldsymbol{s})\|_2 \leq \frac{1}{2}$.
- **HSM:** $J_{\text{hsm}}(s)$ is block diagonal with each block a local softmax Jacobian of spectral norm at most $1/2$, so the overall spectral norm is bounded by $\sup_s \|J_{\text{hsm}}(\boldsymbol{s})\|_2 \leq \frac{1}{2}$.
- **RG:** RG-ALS utilizes Alternating Least Squares rather than gradient descent methods.

The induced condition numbers yield the ordering of linear convergence factors

$$\rho_{\text{NCE}} < \rho_{\text{SSM}} = \rho_{\text{SM}} = \rho_{\text{HSM}}. \tag{76}$$

## B.5 DELTA METHOD

### B.5.1 SETUP

The $\Delta$-method states that if $X_k$ is an average of $k$ i.i.d. random variables with mean $\mu_X$ and variance $\sigma_X^2$, and if $g$ is twice continuously differentiable at $\mu_X$, then

$$g(X_k) \approx g(\mu_X) + g'(\mu_X)(X_k - \mu_X) + \tfrac{1}{2}g''(\mu_X)(X_k - \mu_X)^2, \tag{77}$$

which yields tractable approximations for both the expectation and the variance of $g(X_k)$. This fits our setting naturally: most softmax approximations replace the log-partition $\log \Omega_\star^*(\boldsymbol{s})$ with a smooth transform of a sample average statistic.

**Setup.** We write the approximate conjugate in generic form

$$\Omega^*(\boldsymbol{s};\xi) = g\big(X(\xi)\,;\,\xi\big), \qquad X(\xi) = \frac{1}{k}\sum_{j=1}^{k} h\big(s_{y'_j},\,\xi\big), \tag{78}$$

where the negatives $y'_j$ are drawn i.i.d. from the proposal encoded in $\xi$. Here $g, h$ are scheme-specific but assumed smooth around $\mu_X := \mathbb{E}_\xi[X(\xi)]$. By i.i.d. properties,

$$\mu_X = \mathbb{E}_\xi[X(\xi)], \qquad \sigma_X^2 = \mathrm{Var}_\xi[X(\xi)] = \tfrac{1}{k}\mathrm{Var}_\xi\big(h(s_{y'},\xi)\big). \tag{79}$$

**Bias.** Conditioning on $\boldsymbol{x}$ (hence on $\boldsymbol{s}$), the pointwise bias is

$$\mathrm{Bias} = \mathbb{E}_\xi[\Omega^*(\boldsymbol{s};\xi)] - \Omega_\star^*(\boldsymbol{s}). \tag{80}$$

Applying the second-order Delta expansion of $g(\cdot;\xi)$ at $\mu_X$ gives

$$\mathbb{E}_\xi[\Omega^*(\boldsymbol{s};\xi)] \approx g(\mu_X;\xi) + \tfrac{1}{2}\,g''(\mu_X;\xi)\,\sigma_X^2, \tag{81}$$

so that

$$\boxed{\mathrm{Bias} \approx \underbrace{g(\mu_X;\xi) - \Omega_\star^*(\boldsymbol{s})}_{\text{asymptotic bias}} + \underbrace{\tfrac{1}{2}\,g''(\mu_X;\xi)\,\sigma_X^2}_{\text{curvature bias}}.} \tag{82}$$

**Variance.** Similarly, the Delta method yields

$$\boxed{\mathrm{Var} := \mathrm{Var}_\xi(\Omega^*(\boldsymbol{s};\xi)) \approx [g'(\mu_X;\xi)]^2\sigma_X^2 = \frac{1}{k}[g'(\mu_X;\xi)]^2\,\mathrm{Var}_\xi\big(h(s_{y'},\xi)\big).} \tag{83}$$

### B.5.2 DECOMPOSITION FOR REPRESENTATIVE APPROXIMATIONS

(I) SAMPLED SOFTMAX - SIMPLE(UNCORRECTED)

$$\Omega_{\text{SSM-Simple}}^*(\boldsymbol{s},\xi) = e^{s_y} + \sum_{i=1}^{k} e^{s_{y'_i}}, \qquad X_k = \frac{1}{k}\sum_{i=1}^{k} e^{s_{y'_i}}, \quad g(z) = \log\big(e^{s_y} + kz\big). \tag{84}$$

so $\mu_X = \mathbb{E}_Q[e^{s_{y'}}]$, $\sigma_X^2 = \frac{1}{k}\mathrm{Var}_Q(e^{s_{y'}})$. $g'(z) = \frac{k}{e^{s_y}+kz}$, $g''(z) = -\frac{k^2}{(e^{s_y}+kz)^2}$.

**Bias.** Let $\hat\Omega^* = e^{s_y} + k\,\mathbb{E}_Q[e^{s_{y'}}]$, then

$$\boxed{\mathrm{Bias}_{\text{SSM-Simple}} = \underbrace{\log\frac{e^{s_y} + k\mathbb{E}_Q[e^{s_{y'}}]}{\sum_i e^{s_{y_i}}}}_{\text{asymptotic}} - \underbrace{\frac{k}{2(\hat\Omega^*)^2}\,\mathrm{Var}_Q(e^{s_{y'}})}_{\text{curvature}}} \tag{85}$$

**Variance.**

$$\boxed{\mathrm{Var}_{\text{SSM-Simple}} = \frac{k}{(\hat\Omega^*)^2}\,\mathrm{Var}_Q(e^{s_{y'}})} \tag{86}$$

(II) SAMPLED SOFTMAX (UNBIASED CORRECTION)

$$\Omega_{\text{SSM}}^*(\boldsymbol{s}, \xi) = \frac{1}{k} \sum_{i=1}^{k} \frac{e^{s_{y_i'}}}{Q(y_i')} = X_k, \quad g(z) = \log z, \tag{87}$$

so $\mu_X = \mathbb{E}[X_k] = \Omega_\star^*(\boldsymbol{s})$ and $\text{Var}(X_k) = \frac{1}{k}\text{Var}_Q\big(\frac{e^s}{Q}\big)$. $g'(z) = 1/z$, $g''(z) = -1/z^2$.

**Bias.**

$$\boxed{\text{Bias}_{\text{SSM}}^{\text{asym}} = g(\mu_X) - \log \Omega_\star^*(\boldsymbol{s}) = 0} \tag{88}$$

$$\boxed{\text{Bias}_{\text{SSM}}^{\text{curv}} = \frac{1}{2}\,g''(\mu_X)\,\text{Var}(X_k) = -\frac{1}{2k}\,\frac{\text{Var}_Q\big(\frac{e^s}{Q}\big)}{\Omega_\star^*(\boldsymbol{s})^2} = -\frac{1}{2k}\,\chi^2(P_{\boldsymbol{s}}\|Q)} \tag{89}$$

where $\chi^2(P_{\boldsymbol{s}}\|Q)$ is the $\chi^2$–divergence between the target softmax distribution $P_{\boldsymbol{s}}$ and the proposal $Q$. Hence $\text{Bias}_{\text{SSM}} = -\frac{1}{2k}\chi^2(P_{\boldsymbol{s}}\|Q)$.

**Variance.**

$$\boxed{\text{Var}_{\text{IS}} = \big(g'(\mu_X)\big)^2\text{Var}(X_k) = \frac{1}{k}\,\frac{\text{Var}_Q\big(\frac{e^s}{Q}\big)}{\Omega_\star^*(\boldsymbol{s})^2} = \frac{1}{k}\,\chi^2(P_{\boldsymbol{s}}\|Q)} \tag{90}$$

(III) NOISE-CONTRASTIVE ESTIMATION (NCE)

Let $y^+ \sim P_{\boldsymbol{s}}$ and $y_1', \ldots, y_k' \overset{i.i.d.}{\sim} Q$, define the mixture distribution $M_k = \frac{P_s + kQ}{1+k}$ and

$$\psi(j) := \log \frac{Q(j)}{M_k(j)} = -\log\big(k + t(j)\big), \qquad t(j) := \frac{P_{\boldsymbol{s}}(j)}{Q(j)}. \tag{91}$$

Collect the negative samples via the empirical mean

$$X_k := \frac{1}{k}\sum_{i=1}^{k}\psi(y_i'), \quad \mu_X = \mathbb{E}_Q[\psi(y')], \quad \sigma_X^2 = \text{Var}(X_k) = \frac{1}{k}\text{Var}_Q\big(\psi(y')\big). \tag{92}$$

Then the NCE objective can be written as

$$\ell_{\text{NCE}}(y^+, \{y_i'\}; \boldsymbol{s}) = \underbrace{\log \frac{P_{\boldsymbol{s}}(y^+)}{M_k(y^+)}}_{\text{no randomness}} + \underbrace{k\,X_k}_{g(X_k)} + \text{const}, \tag{93}$$

which fits the template by taking

$$\Omega_{\text{NCE}}^*(\boldsymbol{s}; \xi) = g\big(X_k; \boldsymbol{s}\big), \qquad g(z; \boldsymbol{s}) = \log \frac{P_{\boldsymbol{s}}(y^+)}{M_k(y^+)} + k\,z + \text{const}. \tag{94}$$

Note that $g'(z) = k$ and $g''(z) = 0$. Conditioning on $\boldsymbol{s}$,

$$\text{Bias}_{\text{NCE}}^{\text{curv}} \approx \frac{1}{2}\,g''(\mu_X)\,\sigma_X^2 = 0, \text{Var}_{\text{NCE}} \approx [g'(\mu_X)]^2\,\sigma_X^2 = k^2 \cdot \frac{1}{k}\text{Var}_Q(\psi) = k\,\text{Var}_Q(\psi). \tag{95}$$

A first-order linearization of $\psi(j) = -\log(k + t(j))$ at $\mathbb{E}_Q[t] = 1$ yields

$$\text{Var}_Q(\psi) \approx \frac{1}{(1+k)^2}\,\chi^2\big(P_{\boldsymbol{s}} \| Q\big) \quad \Rightarrow \quad \boxed{\text{Var}_{\text{NCE}} \approx \frac{k}{(1+k)^2}\,\chi^2\big(P_{\boldsymbol{s}} \| Q\big) \sim \frac{1}{k}\,\chi^2}. \tag{96}$$

Taking expectation over $y^+ \sim P_{\boldsymbol{s}}$ and $y' \sim Q$ gives

$$\mathbb{E}\big[\ell_{\text{NCE}}\big] = -\Big(\text{KL}(P_{\boldsymbol{s}}\|M_k) + k\,\text{KL}(Q\|M_k)\Big) + \text{const}$$
$$= -(1+k)\,\text{JS}_\tau(P_{\boldsymbol{s}}\|Q) + \text{const}, \quad \tau = \frac{1}{1+k}. \tag{97}$$

Hence, relative to the exact softmax conjugate $\log \sum_j e^{s_j}$, NCE exhibits a structural bias governed by $\text{JS}_\tau(P_{\boldsymbol{s}}\|Q)$:

$$\boxed{\text{Bias}_{\text{NCE}}^{\text{asym}} \propto (1+k)\,\text{JS}_\tau(P_{\boldsymbol{s}}\|Q), \tau = \frac{1}{1+k}} \tag{98}$$

(VI) HIERARCHICAL SOFTMAX (HSM)

Hierarchical Softmax replaces the flat softmax distribution

$$P_{\boldsymbol{s}}(y) \;=\; \frac{e^{s_y}}{\sum_{j=1}^{C} e^{s_j}} \tag{99}$$

with a tree-structured factorization. Each class $y$ is uniquely represented by a path $(n_1, \ldots, n_{L_y})$ from the root to a leaf, where each node $n_\ell$ has an associated binary classifier with score $s_{n_\ell}$. The hierarchical probability is

$$P_{\mathrm{HSM}}(y \mid \boldsymbol{s}) \;=\; \prod_{\ell=1}^{L_y} \sigma\big(b_{n_\ell} \cdot s_{n_\ell}\big), \tag{100}$$

where $b_{n_\ell} \in \{\pm 1\}$ indicates the branch direction.

**Bias.** Since HSM is deterministic (no sampling), the curvature bias vanishes:

$$\boxed{\mathrm{Bias}_{\mathrm{HSM}}^{\mathrm{curv}} = 0} \tag{101}$$

The asymptotic bias is the pointwise gap between the hierarchical surrogate and the exact softmax conjugate:

$$\boxed{\mathrm{Bias}_{\mathrm{HSM}}^{\mathrm{asym}} = \Omega_{\mathrm{HSM}}^*(\boldsymbol{s}) - \Omega_\star^*(\boldsymbol{s}) = -\log P_{\mathrm{HSM}}(y \mid \boldsymbol{s}) - \log\Big(\frac{\exp s_y}{\sum_j \exp s_j}\Big).} \tag{102}$$

Taking expectation over $y \sim P_{\boldsymbol{s}}$ yields

$$\mathbb{E}_{y \sim P_{\boldsymbol{s}}}\big[\mathrm{Bias}_{\mathrm{HSM}}^{\mathrm{asym}}\big] = \mathrm{KL}\big(P_{\boldsymbol{s}} \,\|\, P_{\mathrm{HSM}}\big), \tag{103}$$

which quantifies how the tree factorization departs from the flat softmax distribution.

**Variance.** Again, since HSM is deterministic (no randomness in $\xi$),

$$\boxed{\mathrm{Var}_{\mathrm{HSM}} = 0.} \tag{104}$$

(V) RG LOSS (QUADRATIC TAYLOR SURROGATE)

The conjugate is generated from the Taylor expansion of $\log \sum_{j=1}^{C} e^{s_j}$ at $\boldsymbol{0}$:

$$\Omega_{\mathrm{RG}}^*(\boldsymbol{s}) = \log C + \frac{1}{C} \sum_{j=1}^{C} s_j + \frac{1}{2} \boldsymbol{s}^\top\Big(\frac{1}{C}I - \frac{1}{C^2}\boldsymbol{1}\boldsymbol{1}^\top\Big)\boldsymbol{s}. \tag{105}$$

**Bias.** This surrogate is deterministic (no sampling), hence

$$\boxed{\mathrm{Bias}_{\mathrm{RG}}^{\mathrm{asym}} = \Omega_{\mathrm{RG}}^*(\boldsymbol{s}) - \log \sum_j e^{s_j} = O(\|\boldsymbol{s}\|^3) \qquad \mathrm{Bias}_{\mathrm{RG}}^{\mathrm{curv}} = 0} \tag{106}$$

**Variance.**

$$\boxed{\mathrm{Var}_{\mathrm{RG}^2} = 0} \tag{107}$$

### B.5.3 ANALYSIS ON ASYMPTOTIC BIAS RESULTS

Building on the Delta framework, we compare the loss-level bias and sampling variance across approximations relative to softmax MLE. Among all, SSM is the closest to MLE: it is unbiased at the loss level and its curvature error decays as $O(k^{-1})$ with coefficients governed by the proposal mismatch $\chi^2(P_{\boldsymbol{s}}\|Q)$. By contrast, the remaining methods exhibit a non-vanishing structural bias with respect to the exact log-partition, which can translate into systematic training differences even when variance is small:

**SSM-Simple.** Method-induced bias from replacing $\log\sum_j e^{s_j}$ by $\log\big(e^{s_y} + k\,\mathbb{E}_Q e^{s_{y'}}\big)$; still enjoys $O(k^{-1})$ sampling variance, but the structural gap does not vanish with $k$.

**NCE.** Optimizes a contrastive proxy $-(1+k)\,\mathrm{JS}_{1/(1+k)}(P_{\boldsymbol{s}}\|Q)$ rather than the softmax conjugate, hence exhibits a structural loss-level bias. Nevertheless, Gutmann & Hyvärinen (2010) shows that NCE shares the MLE stationary point and the gradient-level discrepancy shrinks as $k$ increases; its sampling variance also scales as $O(k^{-1})$ via a $\chi^2$–type coefficient.

**HSM.** Deterministic hierarchical factorization induces a bias quantified by $\mathrm{KL}(P_{\boldsymbol{s}}\|P_{\mathrm{HSM}})$; the structural gap is independent of $k$.

**RG.** A deterministic quadratic surrogate with irreducible approximation bias $O(\|\boldsymbol{s}\|^3)$; while it can preserve consistency properties (Pu et al., 2025), within the bias–variance view the structural bias remains.

Among softmax-family approximations, only SSM eliminates loss-level bias relative to MLE; all others retain a structural gap that may impact performance depending on the proposal $Q$, model capacity, and class hardness. Besides, non-sampling surrogates incur neither curvature bias nor sampling variance, yielding fully deterministic training signals. When the structural discrepancy $\mathrm{Bias}^{\mathrm{asym}}$ is small, these methods has the potential on exhibiting stable optimization and competitive accuracy.

## C  Experimental Results

### C.1  Settings

#### C.1.1  Dataset and Evaluation

**Dataset.** We evaluate our method on three public datasets: **MovieLens-1M(ML-1M)**, **Gowalla**, and **Amazon-Electronics(Electronics)**, collected from different real-world online platforms. **ML-1M** contains explicit user ratings on movies with a 1–5 scale. **Gowalla** is a location-based social networking service where users share their locations via check-ins. **Electronics** collects customers' reviews and ratings (1–5) on electronics products on the Amazon platform. For **ML-1M** and **Amazon-Electronics**, we treat items rated below 3 as negatives and the remaining ones as positives. We employ the widely used $k$-core filtering strategy to remove users and items with fewer than 10 interactions. The detailed statistics after filtering are shown in Table 5.

Table 5: Statistics of datasets.

| Dataset | #User | #Item | #Interact | Sparsity |
|---|---|---|---|---|
| **ML-1M** | 6,038 | 3,307 | 835,789 | 95.81% |
| **Gowalla** | 29,858 | 40,981 | 1,027,370 | 99.16% |
| **Electronics** | 192,403 | 63,001 | 1,689,188 | 99.99% |

**Backbones**.

**MF** (Rendle et al., 2012): Matrix Factorization is one of the most fundamental and widely adopted two-tower architectures. It learns linear embeddings for users and items to obtain their latent representations. Owing to its simplicity, rapid convergence, and strong scalability, **MF** is often employed as a baseline for benchmarking and for systematically comparing the performance of different loss functions.

**SASRec** (Kang & McAuley, 2018) represents a state-of-the-art sequential recommendation approach that applies self-attention mechanisms to model user–item interaction sequences. By capturing both short- and long-term dependencies, this model is capable of effectively representing dynamic user preferences. Its flexibility in handling variable-length sequences and its strong predictive accuracy for next-item recommendation make **SASRec** particularly suitable for studying the influence of sequential patterns in user behavior.

**LightGCN** (He et al., 2020) is a leading graph-based collaborative filtering model within the two-tower paradigm. It employs simplified graph convolutional networks to learn user and item representations by propagating collaborative signals over the user–item interaction graph. With its efficient

message-passing mechanism, concise design, and strong empirical performance, **LightGCN** serves as a representative benchmark for evaluating the effectiveness of graph-based recommendation techniques.

**Data Split.** For each dataset, we divide the interactions of every user into training, validation, and test subsets with a ratio of $\{0.8, 0.1, 0.1\}$. The validation subset is employed to monitor and tune model performance during training, while the test subset provides the basis for the final comparative evaluation.

### C.1.2 EVALUATION METRICS

Following prior discussion, we adopt both classification-oriented and ranking-oriented metrics to comprehensively evaluate model performance. Specifically, we report **P@k**, **R@k**, and **N@k** with $k = 20$ across all datasets. These metrics jointly capture both classification and ranking accuracy, which are critical for large-scale recommendation scenarios.

### C.1.3 BASELINES

We compare **Softmax-family** surrogate losses discussed in our prior sections:

- **Exact losses: Softmax**, **Sparsemax** (Martins & Astudillo, 2016), $\alpha$-**Entmax** (Peters et al., 2019)($\alpha = 1.5$ as representative), and **Rankmax** (Kong et al., 2020) (with simplest $\Delta_{n,1}$ form).

- **Approximate losses: SSM** (Jean et al., 2015), **NCE** (Gutmann & Hyvärinen, 2010), **HSM** (Morin & Bengio, 2005), and **RG** (Pu et al., 2025).

It is worth highlighting that **HSM** and **RG** differ from other Softmax-family losses in terms of their structural and optimization requirements. Specifically, **HSM** leverages a hierarchical decomposition of the output space through a Huffman tree, which tightly couples the loss with the model architecture. This design makes it difficult to seamlessly integrate **HSM** into diverse backbones such as SASRec or LightGCN. To ensure a fair comparison, we therefore report **HSM** results only under the MF backbone. Similarly, **RG** employs an optimization strategy based on Alternating Least Squares (ALS), rather than the commonly used gradient descents. This reliance on ALS prevents its straightforward adaptation to sequential or graph-based recommenders. Consequently, we restrict the evaluation of **RG** to the MF backbone and present its results exclusively in the corresponding tables for direct comparison.

### C.1.4 IMPLEMENTATION DETAILS

To ensure fair and consistent comparisons, all backbone models are implemented with an identical architectural configuration across different loss functions. The embedding dimension is fixed at 64 for all models. For **SASRec**, we adopt two self-attention blocks with a dropout rate of 0.5, while for **LightGCN**, a two-layer graph convolutional network is employed. All methods are trained on a single NVIDIA RTX-3090 GPU with 24GB of memory.

All models and baselines(except for RG) are trained under aligned protocols to ensure fairness:

- Optimizer: Adam (Kingma & Ba, 2014) with batch size of 2048.

- Learning rate: tuned over $\{1e^{-3}, 5e^{-4}, 1e^{-4}\}$ using validation N@20, results are reported in C.2.
- For sampling-based approximations, we choose $k = 100$ for comparision with exact methods and sweep the number of negative samples $k \in \{5, 10, 50, 100\}$ in further discussions.

### C.2 Q1: ACCURACY UNDER ALIGNED PROTOCOLS

Tables 6 and 7 report the results of different learning rate selections under MF and SASRec, respectively. Notice that RG utilizes ALS optimization, which need not fine-tuning on learning rate.

The results reveal several consistent patterns regarding the behavior of different softmax-family losses under aligned training protocols. The key findings can be summarized as follows:

- *Findings 1*. Softmax supports larger learning rates compared to sparse alternatives. Across datasets, we observe that Softmax achieves its best performance at relatively large learning

Table 6: MF performance under different learning rates. **Bold** denote the best performance of a method under different lr. **Blue** indicate the overall best performance under each data–metric pair.

| Loss | Metric | ML-1M | | | Electronics | | | Gowalla | | |
|---|---|---|---|---|---|---|---|---|---|---|
| | | lr=1e-3 | lr=5e-4 | lr=1e-4 | lr=1e-3 | lr=5e-4 | lr=1e-4 | lr=1e-3 | lr=5e-4 | lr=1e-4 |
| Softmax | N@20 | **0.2661** | 0.2658 | 0.2482 | **0.0235** | 0.0231 | 0.0146 | 0.0947 | **0.0953** | 0.0571 |
| | P@20 | **0.1465** | 0.1462 | 0.1391 | **0.0027** | 0.0027 | 0.0018 | 0.0188 | **0.0190** | 0.0122 |
| | R@20 | **0.2937** | 0.2930 | 0.2596 | **0.0496** | 0.0494 | 0.0333 | **0.1716** | 0.1704 | 0.0970 |
| Sparsemax | N@20 | 0.2353 | **0.2536** | 0.2463 | 0.0214 | **0.0240** | 0.0166 | 0.0782 | 0.0948 | **0.0992** |
| | P@20 | 0.1228 | **0.1345** | 0.1344 | 0.0023 | **0.0027** | 0.0019 | 0.0150 | 0.0180 | **0.0203** |
| | R@20 | 0.2620 | **0.2769** | 0.2651 | 0.0421 | **0.0491** | 0.0347 | 0.1375 | 0.1636 | **0.1739** |
| Entmax-1.5 | N@20 | **0.2643** | 0.2631 | 0.2557 | 0.0244 | **0.0255** | 0.0187 | 0.0704 | 0.0733 | **0.0759** |
| | P@20 | 0.1410 | **0.1426** | 0.1383 | 0.0028 | **0.0029** | 0.0022 | 0.0148 | 0.0156 | **0.0161** |
| | R@20 | **0.2897** | 0.2852 | 0.2778 | 0.0510 | **0.0529** | 0.0410 | 0.1232 | 0.1297 | **0.1353** |
| Rankmax | N@20 | **0.2851** | 0.2835 | 0.2621 | 0.0218 | **0.0225** | 0.0145 | 0.1030 | **0.1031** | 0.0654 |
| | P@20 | **0.1543** | 0.1535 | 0.1452 | 0.0025 | **0.0026** | 0.0018 | 0.0206 | **0.0206** | 0.0141 |
| | R@20 | **0.3005** | 0.2985 | 0.2710 | 0.0461 | **0.0472** | 0.0331 | 0.1790 | **0.1794** | 0.1132 |
| SSM | N@20 | 0.2644 | **0.2645** | 0.2465 | 0.0236 | **0.0246** | 0.0147 | **0.0848** | 0.0816 | 0.0539 |
| | P@20 | **0.1462** | 0.1461 | 0.1388 | 0.0028 | **0.0028** | 0.0018 | **0.0173** | 0.0169 | 0.0116 |
| | R@20 | 0.2908 | **0.2910** | 0.2569 | 0.0509 | **0.0525** | 0.0336 | **0.1522** | 0.1458 | 0.0955 |
| NCE | N@20 | **0.2423** | 0.2420 | 0.2227 | **0.0238** | 0.0232 | 0.0150 | **0.0518** | 0.0514 | 0.0451 |
| | P@20 | **0.1374** | 0.1364 | 0.1280 | **0.0029** | 0.0028 | 0.0018 | **0.0110** | 0.0109 | 0.0094 |
| | R@20 | 0.2661 | **0.2687** | 0.2283 | **0.0528** | 0.0520 | 0.0339 | **0.0903** | 0.0899 | 0.0794 |
| HSM | N@20 | 0.1120 | 0.1193 | **0.1202** | **0.0134** | 0.0129 | 0.0123 | **0.0512** | 0.0496 | 0.0464 |
| | P@20 | 0.0739 | 0.0736 | **0.0742** | **0.0016** | 0.0015 | 0.0014 | **0.0114** | 0.0111 | 0.0104 |
| | R@20 | **0.1497** | 0.1236 | 0.1249 | **0.0297** | 0.0283 | 0.0267 | **0.0938** | 0.0915 | 0.0845 |
| RG | N@20 | | **0.2785** | | | **0.0274** | | | **0.0894** | |
| | P@20 | | **0.1521** | | | **0.0030** | | | **0.0200** | |
| | R@20 | | **0.2945** | | | **0.0563** | | | **0.1476** | |

Table 7: SASRec results under different learning rates. **Bold** denote the best performance of a method under different lr. **Blue** indicate the overall best performance under each data–metric pair.

| Loss | Metric | ML-1M | | | Electronics | | | Gowalla | | |
|---|---|---|---|---|---|---|---|---|---|---|
| | | lr=1e-3 | lr=5e-4 | lr=1e-4 | lr=1e-3 | lr=5e-4 | lr=1e-4 | lr=1e-3 | lr=5e-4 | lr=1e-4 |
| Softmax | N@20 | **0.1647** | 0.1643 | 0.1630 | **0.0355** | 0.0351 | 0.0348 | 0.0459 | **0.0470** | 0.0462 |
| | P@20 | **0.0180** | 0.0178 | 0.0177 | **0.0036** | 0.0036 | 0.0036 | 0.0048 | **0.0049** | 0.0049 |
| | R@20 | **0.3599** | 0.3566 | 0.3546 | **0.0728** | 0.0727 | 0.0723 | 0.0968 | **0.0988** | 0.0977 |
| Sparsemax | N@20 | 0.1480 | **0.1490** | 0.0229 | 0.0170 | **0.0246** | 0.0117 | 0.0338 | 0.0347 | **0.0351** |
| | P@20 | 0.0158 | **0.0160** | 0.0027 | 0.0020 | **0.0027** | 0.0014 | 0.0037 | 0.0038 | **0.0038** |
| | R@20 | 0.3165 | **0.3198** | 0.0542 | 0.0399 | **0.0540** | 0.0276 | 0.0746 | 0.0754 | **0.0760** |
| Entmax-1.5 | N@20 | 0.0952 | 0.0932 | **0.0972** | 0.0171 | **0.0172** | 0.0167 | 0.0154 | **0.0154** | 0.0153 |
| | P@20 | 0.0111 | 0.0108 | **0.0113** | 0.0020 | **0.0020** | 0.0019 | 0.0017 | 0.0017 | **0.0017** |
| | R@20 | 0.2229 | 0.2158 | **0.2264** | 0.0398 | **0.0407** | 0.0389 | 0.0337 | 0.0336 | **0.0338** |
| Rankmax | N@20 | 0.1753 | **0.1789** | 0.1729 | **0.0357** | 0.0353 | 0.0344 | 0.0469 | 0.0467 | **0.0474** |
| | P@20 | 0.0185 | **0.0188** | 0.0183 | **0.0036** | 0.0036 | 0.0035 | 0.0049 | 0.0049 | **0.0050** |
| | R@20 | 0.3695 | **0.3756** | 0.3667 | **0.0721** | 0.0716 | 0.0694 | 0.0984 | 0.0983 | **0.0993** |
| SSM | N@20 | **0.1524** | 0.1501 | 0.1486 | 0.0251 | **0.0260** | 0.0134 | 0.0394 | **0.0398** | 0.0394 |
| | P@20 | **0.0171** | 0.0169 | 0.0166 | 0.0028 | **0.0029** | 0.0017 | 0.0044 | **0.0044** | 0.0044 |
| | R@20 | **0.3428** | 0.3385 | 0.3319 | 0.0567 | **0.0576** | 0.0331 | 0.0872 | **0.0873** | 0.0873 |
| NCE | N@20 | 0.1274 | **0.1325** | 0.1281 | 0.0173 | 0.0171 | **0.0175** | **0.0309** | 0.0304 | 0.0262 |
| | P@20 | 0.0153 | **0.0158** | 0.0151 | 0.0020 | 0.0019 | **0.0020** | **0.0035** | 0.0034 | 0.0029 |
| | R@20 | 0.3062 | **0.3158** | 0.3029 | 0.0394 | 0.0384 | **0.0399** | **0.0702** | 0.0684 | 0.0581 |

rates (e.g., $1e-3$ for ML-1M and Electronics, $5e-4$ for Gowalla), whereas sparse variants (Sparsemax, Entmax-1.5, and Rankmax) require smaller step sizes to remain stable. This empirical evidence aligns closely with our theoretical analysis in the earlier sections: the smaller Jacobian spectral norm of Softmax leads to smoother optimization dynamics, thereby allowing more aggressive learning rates without sacrificing stability. Similar phenomena have also been reported in NLP, where sparse methods were found to require substantially smaller learning rates to achieve convergence (Peters et al., 2019; Correia et al., 2019).

- *Findings 2.* Rankmax validates the importance of hard negatives. Across datasets, Rankmax achieves top performance, especially on ML-1M and Gowalla, where the ranking metrics are clearly superior. This effectiveness can be attributed to its explicit emphasis on hard negative samples, which carry the most informative training signal. By amplifying the contribution of these challenging cases, Rankmax guides the model toward more discriminative representations. These observations are in line with our earlier discussion, where we argued that prioritizing hard negatives is crucial for overcoming the limitations of uniformly weighted objectives.

- *Findings 3.* SSM outperforms NCE in stability. Under same configurations, SSM consistently surpasses NCE. This advantage stems from the more stable bias introduced by SSM, whereas NCE suffers from an irreducible bias that destabilizes the optimization process. The superior reliability of SSM suggests that it provides a steadier and more effective learning signal across datasets.

- *Findings 4.* Sparse methods face limitations in complex architectures. Sparsemax and Entmax-1.5 can perform on par with or even better than Softmax under simple models like MF. However, when deployed in more expressive architectures like SASRec, their lossy compression of training signal becomes critical, leading to large performance drops. An additional experiment on LightGCN with ML-1M (Table 8) further confirms this limitation: due to the LightGCN's high sensitivity to negative signals, all sparse methods, even Rankmax, struggle to converge effectively and provide a huge performance gap. These findings support our conclusion that the WOP property induces an information compression effect that harms optimization in complex scenarios.

- *Findings 5.* Non-sampling approximation methods diverge sharply in performance. We observe that HSM performs poorly across all tasks. This deficiency is likely due to a combination of three factors: (1) the modeling mismatch between HSM and MF, which weakens its representation learning; (2) the non-consistency of its objective with the evaluation metrics, leading to an inherent optimization gap; and (3) the discrepancy between the predefined Huffman tree distribution and the target Softmax-MLE distribution, which introduces substantial bias. In contrast, RG avoids these pitfalls: its modeling design aligns closely with MF, its objective is consistent with the evaluation metrics, and its bias depends only on higher-order terms of the scoring vector, which has negligible impact in practical optimization. As a result, RG consistently demonstrates superior performance across all datasets, validating its effectiveness as a non-sampling approximation to Softmax.

Table 8: LightGCN results on ML-1M under different learning rates. **Bold** denotes the best performance of each method across learning rates.

| Loss | N@20 | | | P@20 | | | R@20 | | |
|---|---|---|---|---|---|---|---|---|---|
| | lr=1e-3 | lr=5e-4 | lr=1e-4 | lr=1e-3 | lr=5e-4 | lr=1e-4 | lr=1e-3 | lr=5e-4 | lr=1e-4 |
| Softmax | **0.2279** | 0.2150 | 0.1268 | **0.1316** | 0.1239 | 0.0740 | **0.2488** | 0.2194 | 0.1281 |
| Sparsemax | **0.1695** | 0.1496 | 0.1226 | **0.0983** | 0.0881 | 0.0748 | **0.1755** | 0.1544 | 0.1278 |
| Entmax-1.5 | 0.1208 | 0.1207 | **0.1212** | 0.0731 | 0.0731 | **0.0734** | 0.1272 | 0.1272 | **0.1280** |
| Rankmax | 0.1208 | 0.1208 | **0.1212** | 0.0731 | 0.0732 | **0.0734** | 0.1273 | 0.1273 | **0.1277** |
| SSM | **0.2358** | 0.2155 | 0.1281 | **0.1339** | 0.1242 | 0.0738 | **0.2523** | 0.2234 | 0.1281 |

Overall, these experiments demonstrate that Softmax tolerates larger learning rates, Rankmax leverages hard negatives to achieve strong performance, and SSM offers more stable optimization than NCE. In contrast, sparse alternatives such as Sparsemax and Entmax struggle in complex architectures due to lossy information compression. Furthermore, non-sampling approximation meth-

ods diverge sharply. Taken together, these five findings align closely with our theoretical analysis, highlighting clear trade-offs between smoothness, stability, and sparsity in normalization losses for recommendation.

## C.3 Q2: Bias-variance decomposition

Our second set of experiments directly examines the bias–variance trade-offs predicted by our theoretical analysis. The results are summarized in Tables 9, 10, 11, and 12. We highlight two key findings:

Table 9: MF results under different negative sample sizes $k$. **Bold** denotes the best performance of each method across $k$ within each dataset.

| Loss | N@20 | | | | P@20 | | | | R@20 | | | |
|------|------|------|------|------|------|------|------|------|------|------|------|------|
| | $k$=5 | $k$=10 | $k$=50 | $k$=100 | $k$=5 | $k$=10 | $k$=50 | $k$=100 | $k$=5 | $k$=10 | $k$=50 | $k$=100 |
| **ML-1M** | | | | | | | | | | | | |
| SSM | 0.2504 | 0.2554 | 0.2630 | **0.2644** | 0.1406 | 0.1430 | 0.1452 | **0.1462** | 0.2738 | 0.2821 | 0.2905 | **0.2908** |
| NCE | 0.2403 | 0.2410 | 0.2409 | **0.2423** | 0.1360 | 0.1360 | 0.1362 | **0.1374** | 0.2643 | 0.2650 | **0.2681** | 0.2661 |
| **Electronics** | | | | | | | | | | | | |
| SSM | 0.0235 | 0.0237 | 0.0245 | **0.0246** | 0.0028 | 0.0028 | 0.0028 | **0.0028** | 0.0514 | 0.0516 | 0.0520 | **0.0525** |
| NCE | 0.0232 | 0.0235 | 0.0235 | **0.0238** | 0.0028 | 0.0028 | **0.0029** | 0.0029 | 0.0523 | 0.0525 | **0.0530** | 0.0528 |
| **Gowalla** | | | | | | | | | | | | |
| SSM | 0.0611 | 0.0675 | 0.0806 | **0.0848** | 0.0129 | 0.0143 | 0.0167 | **0.0173** | 0.1074 | 0.1197 | 0.1450 | **0.1522** |
| NCE | 0.0514 | 0.0515 | 0.0516 | **0.0518** | 0.0109 | 0.0110 | 0.0109 | **0.0110** | 0.0893 | 0.0900 | 0.0899 | **0.0903** |

Table 10: SASRec results under different negative sample sizes $k$. **Bold** denotes the best performance of each method across $k$ within each dataset.

| Loss | N@20 | | | | P@20 | | | | R@20 | | | |
|------|------|------|------|------|------|------|------|------|------|------|------|------|
| | $k$=5 | $k$=10 | $k$=50 | $k$=100 | $k$=5 | $k$=10 | $k$=50 | $k$=100 | $k$=5 | $k$=10 | $k$=50 | $k$=100 |
| **ML-1M** | | | | | | | | | | | | |
| SSM | 0.0179 | 0.1248 | 0.1400 | **0.1524** | 0.0024 | 0.0149 | 0.0160 | **0.0171** | 0.0480 | 0.2978 | 0.3203 | **0.3428** |
| NCE | 0.0174 | 0.1024 | 0.1291 | **0.1325** | 0.0024 | 0.0124 | 0.0153 | **0.0158** | 0.0474 | 0.2473 | 0.3064 | **0.3158** |
| **Electronics** | | | | | | | | | | | | |
| SSM | 0.0148 | 0.0170 | 0.0231 | **0.0260** | 0.0016 | 0.0017 | 0.0024 | **0.0029** | 0.0320 | 0.0344 | 0.0485 | **0.0576** |
| NCE | 0.0157 | 0.0158 | **0.0214** | 0.0175 | 0.0017 | 0.0017 | **0.0023** | 0.0020 | 0.0340 | 0.0342 | **0.0464** | 0.0399 |
| **Gowalla** | | | | | | | | | | | | |
| SSM | 0.0301 | 0.0308 | 0.0368 | **0.0398** | 0.0033 | 0.0034 | 0.0041 | **0.0044** | 0.0668 | 0.0689 | 0.0824 | **0.0873** |
| NCE | 0.0270 | 0.0261 | 0.0292 | **0.0309** | 0.0029 | 0.0028 | 0.0030 | **0.0035** | 0.0584 | 0.0561 | 0.0595 | **0.0702** |

- *Findings 6.* Increasing the negative sample size $k$ yields monotonic improvements for SSM but has negligible effect on NCE. As shown in Tables 9 and 10, SSM exhibits steady gains in N@20, P@20, and R@20 as $k$ grows, whereas NCE remains largely flat across different $k$. This observation aligns closely with our bias–variance decomposition: SSM's bias decreases linearly with larger $k$, while NCE's bias remains fixed and only its variance reduces, leading to limited overall improvement.

- *Findings 7.* Sampling distributions that better approximate the Softmax-MLE distribution significantly improve both SSM and NCE. As reported in Tables 11 and 12, switching from uniform sampling to DNS consistently enhances all metrics across datasets and models. This finding is in line with our theoretical results showing that the bias of both methods depends on the divergence between the proposal distribution and the target Softmax-MLE, and that reducing this divergence directly translates into improved empirical performance.

Together, these findings further corroborate our theoretical framework: the empirical behaviors of SSM and NCE with respect to sample size $k$ and proposal distribution $Q$ precisely match the bias–variance decomposition established in Appendix. B.5.

Table 11: MF results under different proposal distributions $Q$ (DNS draws Top-100 from $k = 500$ candidates). **Bold** denotes the best performance of each method across proposal distributions within each dataset.

| Loss | N@20 | | P@20 | | R@20 | |
|---|---|---|---|---|---|---|
| | Uniform | DNS | Uniform | DNS | Uniform | DNS |
| **ML-1M** | | | | | | |
| SSM | 0.2644 | **0.2739** | 0.1462 | **0.1478** | 0.2908 | **0.2991** |
| NCE | 0.2423 | **0.2769** | 0.1374 | **0.1498** | 0.2661 | **0.3002** |
| **Amazon-Electronics** | | | | | | |
| SSM | 0.0246 | **0.0255** | 0.0028 | **0.0029** | 0.0525 | **0.0536** |
| NCE | 0.0238 | **0.0273** | 0.0029 | **0.0032** | 0.0528 | **0.0599** |
| **Gowalla** | | | | | | |
| SSM | 0.0848 | **0.0908** | 0.0173 | **0.0180** | 0.1522 | **0.1651** |
| NCE | 0.0514 | **0.0599** | 0.0110 | **0.0129** | 0.0903 | **0.1048** |

Table 12: SASRec results under different proposal distributions $Q$ (DNS draws Top-100 from $k = 500$ candidates). **Bold** denotes the best performance of each method across proposal distributions within each dataset.

| Loss | N@20 | | P@20 | | R@20 | |
|---|---|---|---|---|---|---|
| | Uniform | DNS | Uniform | DNS | Uniform | DNS |
| **ML-1M** | | | | | | |
| SSM | 0.1524 | **0.1570** | 0.0171 | **0.0176** | 0.3428 | **0.3497** |
| NCE | 0.1325 | **0.1457** | 0.0158 | **0.0173** | 0.3158 | **0.3474** |
| **Amazon-Electronics** | | | | | | |
| SSM | 0.0260 | **0.0265** | 0.0029 | **0.0030** | 0.0576 | **0.0588** |
| NCE | 0.0175 | **0.0201** | 0.0020 | **0.0024** | 0.0399 | **0.0459** |
| **Gowalla** | | | | | | |
| SSM | 0.0398 | **0.0418** | 0.0044 | **0.0045** | 0.0873 | **0.0895** |
| NCE | 0.0309 | **0.0387** | 0.0035 | **0.0042** | 0.0702 | **0.0836** |

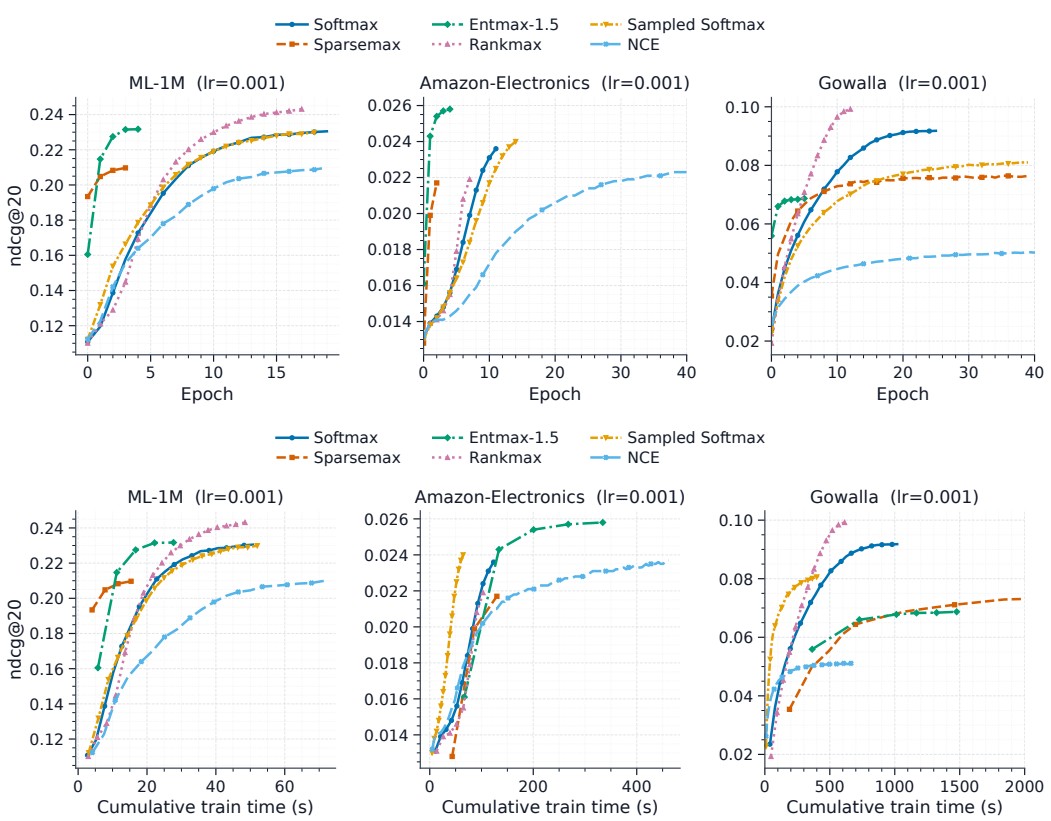

Figure 1: MF backbone: NDCG@20 curves with respect to (top) training epochs and (bottom) cumulative wall-clock time.

## C.4 Q3: TRAINING DYNAMICS AND EFFICIENCY

We further examine the optimization dynamics of different normalization losses by plotting NDCG@20 against both training epochs and cumulative wall-clock time. The results are shown in Figures 1 and 2. The findings are:

- *Findings 8.* Sparse methods converge in very few epochs under simple MF but lose this advantage in complex architectures of SASRec. As illustrated by the NDCG@20-vs.-epoch curves in Figures 1 and 2, sparse variants such as Sparsemax and Entmax quickly concentrate their gradients on a small number of samples, leading to rapid convergence within a handful of epochs for MF. However, this property does not hold for more expressive models like SASRec, where the training dynamics become less favorable and convergence is significantly slower.

- *Findings 9.* Sparse methods incur substantial computational overhead, while sampling-based methods sometimes achieve faster effective training. The NDCG@20-vs.-time curves in Figures 1 and 2 reveal that sparse methods demand heavy computation per iteration, severely hampering their efficiency in practice—this effect is particularly pronounced on large datasets such as Gowalla. By contrast, sampling-based approaches, although limited by a lower performance ceiling, may benefit from much faster iteration speed, and thus reach competitive results more quickly especially in MF backbone.

Together, these findings reinforce our theoretical analysis: sparse objectives compress training signals in ways that distort optimization dynamics, and their high per-iteration cost undermines their practical utility compared to lightweight sampling-based alternatives.

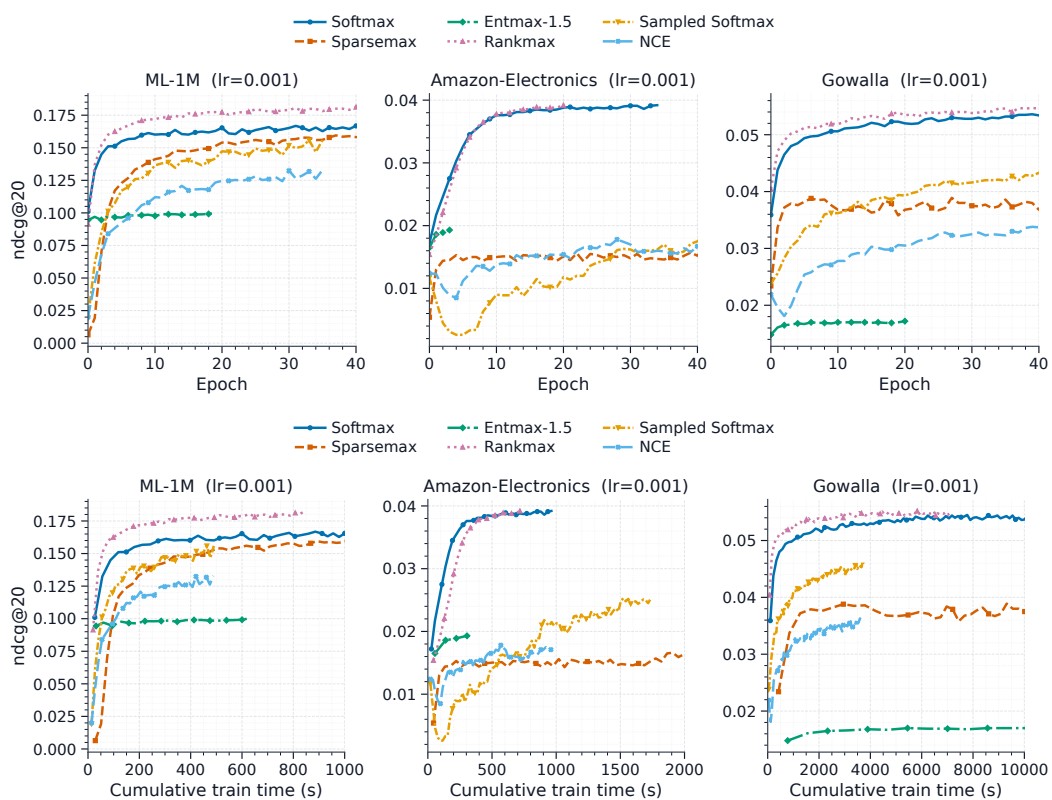

Figure 2: SASRec backbone: NDCG@20 curves with respect to (top) training epochs and (bottom) cumulative wall-clock time.

## C.5 GRADIENT-MAGNITUDE DIAGNOSTICS

As shown in Figure 3, the gradients of WOP losses exhibit a striking sparsity pattern: large portions of the Jacobian are exactly zero (highlighted as black in the heatmaps). This empirical observation is consistent with our theoretical analysis in earlier sections, where we proved that the WOP property inevitably induces information compression by forcing many coordinates to vanish in the gradient. Such sparsity reduces the diversity of training signals available to the model, which in turn explains the limited optimization dynamics observed in our experiments.

## D DISCUSSIONS ON TOP-K OPTIMIZATION

Based on the theoretical and empirical results, we provide a further discussion on two critical aspects regarding the practical optimization of Softmax-based losses: the challenges of bias correction in mini-batch sampling and the discrepancy between calibration and ranking metrics.

**Bias Correction and Compositional Optimization.** SSM may exhibit high curvature bias and variance when the negative sampling size is small. Recent work by Qiu et al. (2022) addresses a similar challenge in the mini-batch training process of NDCG-surrogates by applying compositional optimization to handle the Jensen-type bias. However, there is a structural distinction between the two scenarios: the method in Qiu et al. (2022) handles with an outer nonlinear function by maintaining a global statistic. In contrast, the bias of SSM sampling arises from the normalization term $Z(x) = \sum \exp(s_i)$, which varies dynamically with each query. Maintaining query-specific statistics is computationally infeasible at scale. Consequently, developing a small-sample asymptotically unbiased estimator for Softmax normalization remains a non-trivial open problem.

**Gap Between Top-$K$ Calibration and Ranking Metrics.** Although Softmax loss is Top-$K$ calibrated, empirical results often show it under-performing advanced Top-$K$ ranking surrogates (e.g.,

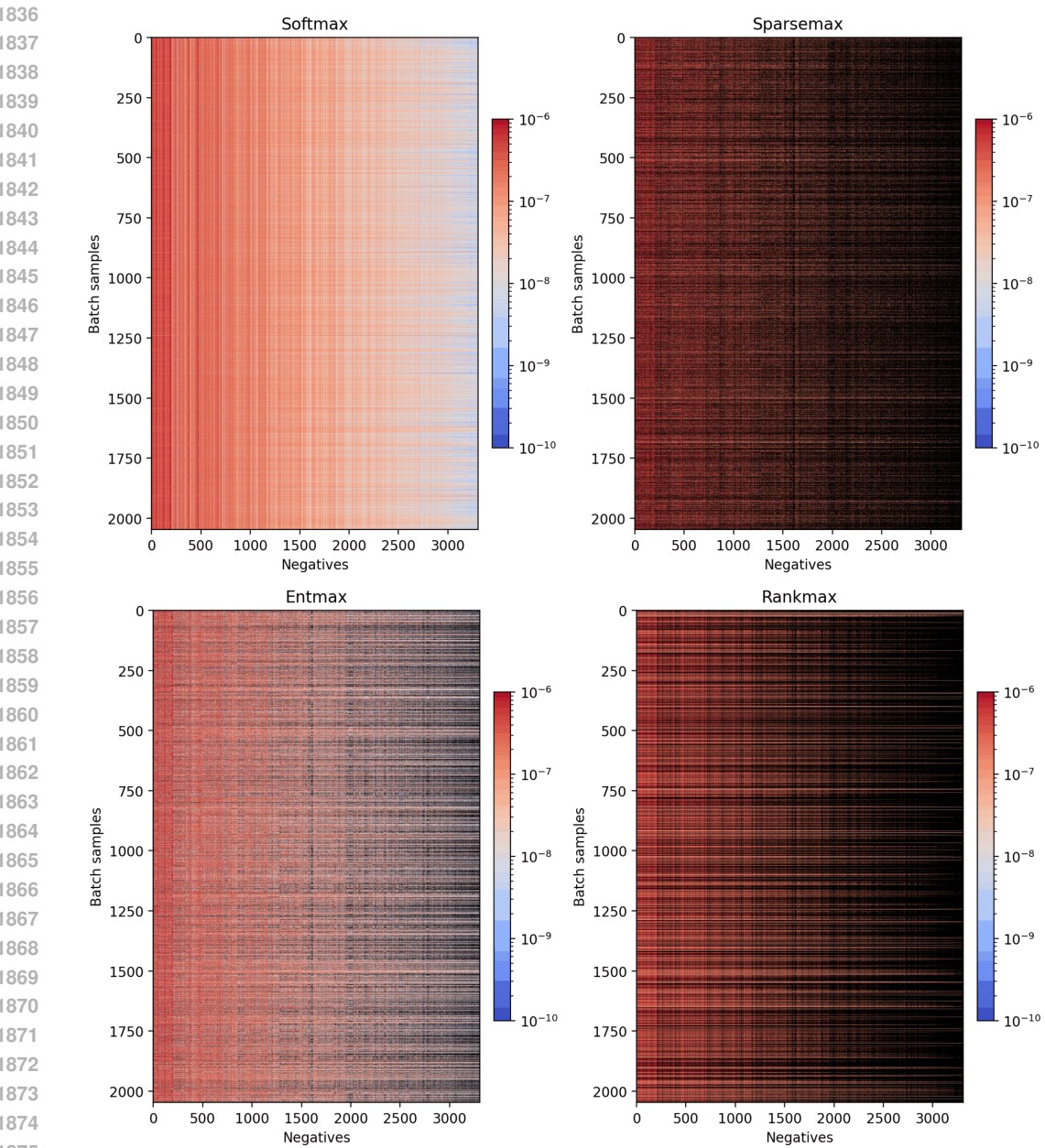

Figure 3: Heatmaps of the gradient (Jacobian) matrices for non-sampling normalization losses. Black regions correspond to zero-valued entries.

Jagerman et al. (2022); Yang et al. (2025)) on metrics like NDCG@$K$. This phenomenon may be explained by the misalignment between calibration targets and ranking objectives. Top-$K$ calibration ensures consistency with metrics such as Recall@$K$ and Precision@$K$ via multilabel reductions (Menon et al., 2019), while cutoffs on ranking metric like NDCG@$K$ shows a different behavior. Softmax loss satisfies consistency with a full-ranking metric NDCG(@$\infty$) at a global length (Bruch et al., 2019; Yang et al., 2024), yet specialized losses (Jagerman et al., 2022; Yang et al., 2025) focus more on the cutoff of NDCG@$K$ metric, thereby achieving better empirical results on top-ranked items. Bridging the theoretical gap between these surrogate losses and position-sensitive metrics remains a promising direction for future research.

# E    LLM USAGE

LLMs were used in the preparation of this manuscript for polishing the writing and checking grammar or style issues. All ideas, theoretical results, and experimental designs were conceived, derived, and implemented by the authors without the assistance of LLMs.

