# OpenReview forum: "Is Softmax Loss all you need? A Principled Analysis of Softmax Loss and its Variants"
_ICLR.cc/2026/Conference — ICLR 2026 Conference Desk Rejected Submission_

### Official Review · Reviewer_RSpv · 2025-10-30

**Soundness:** 4
**Presentation:** 4
**Contribution:** 3
**Rating:** 6
**Confidence:** 4

**Summary:**

This paper systematically analyzes the softmax loss (SM) and its variants, focusing on three theoretical aspects: (1) Consistency: based on the Fenchel-Young framework, the authors compare the consistency of SM with sparse SM variants (including Sparsemax, $\alpha$-Entmax, and Rankmax). (2) Convergence: by estimating the first-order Lipschitz constant, the authors analyze the convergence rates of SM, the sparse variants, and the approximate variants (including SSM, NCE, HSM, RG). (3) Bias and Variance: based on the Delta method, the authors compare the asymptotic bias, curvature bias, and variance of SM and its approximate variants. The theoretical results are validated through extensive experiments.

**Strengths:**

- This paper presents a comprehensive theoretical analysis of the softmax loss and its variants, providing a unified view of widely used softmax-based losses.
- The writing is clear and well-structured, the mathematical derivations are rigorous and easy to follow.
- The experimental results match well with the theoretical findings.

**Weaknesses:**

In general, this paper provides a solid and comprehensive analysis of the softmax losses. I have checked all the main theoretical results and found no major issues (except for possible typos mentioned below). The weaknesses, from my perspective, mainly lie in two aspects:
- This paper does not provide a novel method that improves upon existing softmax variants. However, I believe the theoretical insights provided in this paper are still worth publishing.
- This paper analyzes the general softmax losses, yet it merely conducts experiments on the recommendation tasks. It would be more convincing if the authors could validate their theoretical findings on other classification tasks, e.g., image classification.

**Questions:**

**Questions for Discussion.** I have some further questions mainly for discussion purposes:
- **Improved SSM.** While SSM has a zero asymptotic bias, when the negative sampling size $k$ is small, the variance can be large (as also shown in Table 2). A previous work [R1] improves SSM by simply applying the compositional optimization technique to facilitate small-batch training. While I understand that this method may not be subsumed under the Fenchel-Young framework used in this paper, I wonder if the authors have any insights on comparison between SSM and the proposed method in [R1]?
- **Top-$k$ Calibration.** The top-$k$ calibration property is sufficiently discussed in the prior works and this paper. Nonetheless, in practice, SM (or SSM) is often underperforming the elaborated top-$k$ ranking surrogate losses (e.g., [R2, R3]). Given the theoretical results in this paper, do the authors have any insights on why this is the case?

**Minor Comments.** Here are some minor typos I found in the paper:
- There is a typo in Eq. (23), where $\textbf{p}$ should be $\hat{\textbf{p}}$.
- There exists a typo in the sign of Eq. (98).

**References:**

- [R1] Large-scale Stochastic Optimization of NDCG Surrogates for Deep Learning with Provable Convergence. ICML '22.
- [R2] On Optimizing Top-K Metrics for Neural Ranking Models. SIGIR '22.
- [R3] Breaking the Top-K Barrier: Advancing Top-K Ranking Metrics Optimization in Recommender Systems. KDD '25.

---

> ### Author Response · Authors · 2025-11-20
>
> We sincerely thank Reviewer RSpv for your detailed and thoughtful evaluation. We especially appreciate the positive comments on writing clarity, rigorous mathematical derivations, and the value of providing unified theoretical understandings. We are also grateful for the reviewer's efforts on checking the correctness of our main results and recognized the contribution as solid and worthwhile. Below we will try our best to address the reviewer’s concerns and discussion questions.
>
> **On Weaknesses**
>
> We agree with the reviewer that the contribution of this paper is theoretical rather than proposing new algorithmic designs. Our main attention lies in unifying the isolated landscape of softmax-family losses and provide principled guidelines for choosing among them in large-class learning.  We appreciate that the reviewer acknowledges that these insights remain valuable even without proposing new methods, and we believe further studies would help designing new surrogate losses under this principled framework.
>
> For validations on other scenarios, we fully agree that it is desirable to validate the conclusions beyond recommendation settings (such as on image classifications). At the same time, there are two potential factors that image classification might not be appropriate:
>
> 1. **Ranking metrics.**
>
> Our analysis covers both classification and ranking consistencys. However, image classification tasks provide single-label ground truth and are typically evaluated by Top-k acc, while ranking metrics would be more suitable for multi-label scenarios, which aligns with recommender systems.
>
> 2. **Larger label space where theory is most distinctive.**
>
> Image classification usually involves relatively small label spaces (e.g., CIFAR-100: 100 classes; ImageNet-1K: 1K classes). In contrast, recommendation systems contain tens of thousands to millions of items. The lack-of-gradient-signal issues of WOP mappings become substantially more pronounced as $C$ grows. Thus, the recommendation setting provides a clearer and more discriminative validation of the theoretical differences across the Softmax family.
>
> Nonetheless, we agree that demonstrating generality is important. We have conducted experiments on an initial and classical experiment (CIFAR-100 under ResNet-18), whose results are included below, from which we still can conclude several observations align with our theoretical results. (the choice of learning rate, and the convergence speed of softmax(SOP) is way faster than WOP methods during training) We haven't yet finished but promise to add additional experimental results on larger datasets with larger image classification backbones in our updated manuscript within the discussion period, since my colleagues are also competing for devices during the rebuttal period :)
> ### CIFAR-100 with ResNet-18 backbone within 200 epochs
> | Loss Function | lr | Top-1 Acc (%) | Top-5 Acc (%) |
> |---------------|----|---------------|---------------|
> |   **Softmax**                  | 0.1 | **78.39** | **94.16** |
> |                  | 0.05 | 77.47 | 93.78 |
> |                    | 0.01 | 74.58 | 92.59 |
> |       **Sparsemax**                 | 0.1 | 77.11 | 92.49 |
> | | 0.05 | **77.15** | **92.80** |
> |                        | 0.01 | 70.76 | 88.65 |
> |         **1.5-Entmax**               | 0.1 | 76.82 | 91.80 |
> |  | 0.05 | **77.03** | **92.23** |
> |                        | 0.01 | 72.65 | 89.63 |
> |        **Rankmax**                | 0.1 | 76.23 | 92.02 |
> |   |  0.05 | **76.79** | **92.59** |
> |                       | 0.01 | 73.70 | 91.34 |
> ---

---

> > ### Author Response · Authors · 2025-11-20
> >
> > **Responses to Discussion Questions**
> >
> > **Q1:**
> > Thank you for proposing this perspective! First, we would like to conclude that the NDCG-surrogates in [R1] are objectives with an explicit **outer nonlinearity functions** written as  $L(\theta)=\Phi(\mu(\theta))$ where $\mu(\theta)=\mathbb{E}[g(\xi;\theta)]$, and $\Phi(\cdot)$ is a ranking-sensitive nonlinear function. In such cases, minibatch substitution $\mu(\theta) \leftarrow \hat\mu=\tfrac{1}{B}\sum_b g(\xi_b;\theta)$ induces a Jensen-type bias because $\Phi(\mathbb{E}[g]) \neq \mathbb{E}[\Phi(g)]$.  The method in [R1] specifically addresses this structure by maintaining a global estimate of $\mu(\theta)$ and applying compositional optimization to control the resulting bias and variance.
> >
> > Meanwhile, SM/SSM optimize a sample-averaged loss of the form  $J(\theta)=\frac{1}{N}\sum_{i=1}^N \ell(\theta;x_i,y_i)$, which is a linear average over samples. Therefore, minibatch-SGD
> >
> > $
> > \widehat{\nabla J(\theta)}
> > =\frac{1}{B}\sum_{i\in\mathcal{B}}\nabla_\theta \ell(\theta;x_i,y_i)
> > $
> >
> > is already an unbiased estimator; there is no nonlinear operation in such dimension, and thus no minibatch-level Jensen bias of the kind studied in [R1].
> >
> > In fact, the similarity between [R1] and our framework where softmax losses does exhibit Jensen bias lies in the softmax normalization inside each sample, i.e.,
> >
> > $
> > Z(x,\theta)=\sum_{j=1}^{C}e^{s_j(\theta,x)}
> > =\mathbb{E}_{j\sim Q}\left[\frac{e^{s_j(\theta,x)}}{Q(j)}\right].
> > $
> >
> > Replacing $Z$ with an unbiased estimator $\widehat{Z}$ (through sampling) yields $\mathbb{E}[\log \widehat{Z}] < \log Z,$ a Jensen-type curvature bias, which is carefully discussed in the bias-variance decomposition part of our paper. Besides, such similarity naturally raises another question: Could the compositional technique in [R1] be directly transferred to softmax normalization level? The answer is however, not trivial.
> >
> > In [R1], the inner statistic $\mu(\theta)$ is global and shared across all data, making it feasible to maintain a single running estimate $\mu_t$. In contrast, softmax normalization yields a query-specific expectation $\mu_x(\theta)=\mathbb{E}_{j\sim Q}\left[\frac{e^{s_j(\theta,x)}}{Q(j)}\right], $ which varies dynamically with each sample $x$. Maintaining a separate estimator $\mu_t(x)$ for every query is infeasible at scale, while using a single global $\mu_t$ would change the optimization objective,  thus altering the target rather than correcting the bias.
> >
> > For these structural reasons, directly applying the compositional optimization method of [R1] to SM/SSM’s softmax normalization would require additional modeling and algorithmic design, which might be a potential research direction?
> >
> > **Q2:**
> > Thank you for this important question! This phenomenon is indeed something we also observed in practice and attempted to analyze. Our current understanding is based on the following reasons. In [R2], the evaluation is only limited to NDCG@$K$; [R3] additionally includes Recall@$K$, however, showing the discrepancy much more pronounced on NDCG@$K$ than on Recall@$K$ across losses. This phenomenon implicitly reflects the gap between Top-k calibrated losses (which align with Recall@$K$ / Precision@$K$ through multi-class reduction by [R4] ) and Top-k ranking metrics such as NDCG@$K$.
> >
> > Softmax loss is known to be Top-k calibrated, and this calibration property ensures Bayes consistency with Recall@$K$/Precision@$K$ for any cutoff $K$. However, such consistency does not automatically transfer to NDCG@$K$ (where Softmax Loss is only DCG-consistent under binarized single-click behavior). As a result, NDCG@K-specific loss designs exhibit better performance than SM on their corresponding metrics in practice.
> >
> > Both [R2] and [R3] adopt loss constructions that more directly align with NDCG@$K$. In contrast, there might be another potential analytical perspective on this issue that we are currently working on: the optimal set relations among Bayes-optimal predictors of metrics like NDCG vs its cutoff version, and whether their optimization directions can be aligned during training. We believe this provides a more principled way to understand such observed empirical phenomenon and plan to extend this theoretical analysis in our future work.
> >
> > [R4] Menon, Aditya K., et al. "Multilabel reductions: what is my loss optimising?." Advances in Neural Information Processing Systems 32 (2019).
> >
> >
> > **Typos**
> >
> > Thank you again for catching these mistakes. We have fixed them in our revised manuscript!
> >
> > Once again, we thank Reviewer RSpv for your careful review, constructive feedback, and recognition of our contribution. Your suggestions significantly improved the clarity and scope of our paper.

---

> > > ### Comment · Reviewer_RSpv · 2025-11-22
> > > **Response to Authors' Rebuttal**
> > >
> > > Thanks for the authors' detailed rebuttal, which addressed my questions. I particularly appreciate the authors' careful reading of the references [R1-R3] I provided, as they represent the latest advances in top-K optimization in RS. I sincerely hope the authors can incorporate a discussion of this aspect in the revised version, especially considering that 1) the optimal loss satisfying the top-K calibrated property is not unique, and that 2) there exists a gap between binary classification metrics (e.g., Recall@K) and ranking metrics (e.g., NDCG@K). I am also glad to see the authors' planned next steps in top-K optimization research, and I look forward to your future work.
> > >
> > > In addition, regarding the authors' rebuttal, I have provided some brief responses below:
> > >
> > > - **Regarding [R1]:** The Jensen gap mentioned by the authors indeed exists and is unavoidable, and maintaining a sum‑of‑exp estimator for each query/user is also inevitable. Although I believe that grouping operations can partially alleviate this issue, they may introduce new bias. Therefore, the small-sample asymptotically unbiased estimation for the softmax loss still has substantial room for further development.
> > > - **Regarding [R2] and [R3]:** My current view is that SL essentially optimizes a full‑ranking metric (e.g., NDCG) rather than the top-K metric NDCG@K [R5, R6]. Since NDCG is essentially equivalent to NDCG@$\infty$, while K is typically small in practice (e.g., 10-20), losses specifically designed for top‑K optimization can significantly improve top‑K performance. In particular, both NDCG@K and Recall@K can be improved in practice. However, these losses are often derived by applying certain upper-bound approximations to the metrics, and whether the resulting gap is reasonable remains an open question for further study.
> > >
> > > Finally, I have raised my score to 8, and I hope the authors can further refine their manuscript.
> > >
> > > **References.**
> > >
> > > [R5] An analysis of the softmax cross entropy loss for learning-to-rank with binary relevance. SIGIR '19.
> > >
> > > [R6] PSL: Rethinking and Improving Softmax Loss from Pairwise Perspective for Recommendation. NIPS '24.

---

> ### Author Response · Authors · 2025-11-22
>
> Dear Reviewer RSpv,
>
> Thank you very much for your prompt response! We are pleased that we were able to address your questions, and we are happy to further revise our manuscript, especially for adding discussions on recent research related to Top-k metrics as revealed in [R1-R6]. The revised version of the manuscript will be uploaded within the rebuttal period.

---

### Official Review · Reviewer_wSds · 2025-10-30

**Soundness:** 3
**Presentation:** 3
**Contribution:** 3
**Rating:** 6
**Confidence:** 2

**Summary:**

This paper summarizes some properties of softmax loss and its variants. It provides a unified theoretical study of the softmax-family of loss functions, including consistency properties of different F-Y surrogates, gradient dynamics and computational complexity.

**Strengths:**

This paper is well written and easy to follow. It provides a comprehensive summary of various softmax family losses. The introduction of the SOP/WOP and bias–variance decomposition are theoretically elegant, and offers interpretable insights into trade-offs between accuracy and efficiency of softmax family losses.

**Weaknesses:**

I have some doubts about the convergence rate analysis. First, the softmax family losses are not all strongly convex, and the authors’ analysis of the condition number relies on adding an $l_2$ regularization term to the original problem. Moreover, in practice, loss functions are often nonconvex. Could the authors further explain how the analysis in Section 3.2 is more applicable to practical settings?

**Questions:**

See weaknesses.

---

> ### Author Response · Authors · 2025-11-20
>
> We sincerely thank reviewer wSds for your thoughtful assessment of our work. We appreciate the positive comments regarding clarity and readability, the "comprehensive summary" of Softmax-family, and the "theoretical elegance" of introducing new concepts.
>
> Regarding your concern on convergence-rate analysis, we agree that softmax-family losses are not globally strongly convex, and that deep learning objectives are generally non-convex. In Section 3.2 and Appendix B.4, we follow a standard approach in optimization studies by adding an $\ell_2$​ regularization term to ensure strong convexity of the regularized objective, which allows the condition number to be comparable across losses. This treatment is closely aligned with practical training: under standard SGD-type methods, weight decay is mathematically equivalent to $\ell_2$​ regularization. The resulting curvature properties directly reflect real-world optimization. Our analysis reflects local geometry induced by the regularized losses, which is consistent with practical training.

---

> > ### Comment · Reviewer_wSds · 2025-11-23
> >
> > I appreciate the author’s reply. In practical problems, especially in deep learning, even with the of $l_2$ regularization, the objective function is still likely to be non‑convex. Therefore, I still feel that using the condition number directly to measure the curvature of loss function is oversimplified.

---

> ### Author Response · Authors · 2025-11-24
>
> We appreciate reviewer wSds again for your reply, which significantly helps us improve the quality of our manuscript. We agree that our description on the strong convexity assumption seems to be a **simplification** to practical deep networks ignoring non-convex DNN structures, which should be further clarified to readers. A more **precise** statement is that the convexity/strong-convexity we rely on applies to the loss with respect to the prediction head (logits), not to the full deep-network parameters. However, the smoothness of prediction head still largely matter for the whole network's optimization. To avoid confusion, we are glad to supplement the following points into our manuscript:
>
> **(1) Convexity of the Fenchel-Young loss.**
>
> The convexity of the F-Y loss $\mathcal{L}(\textbf{y,s})$ with respect to the score vector $s$ is guaranteed by its construction ([R1], Prop. 2.3). Therefore, the regularized head-objective becomes $λ$-strongly convex if we add a $l_2$ regularizer directly on $s$, and our smoothness/condition-number analysis is valid under such assumption.
>
> **(2) Deep networks are non-convex, even with $l_2$ regularization.**
>
> We fully acknowledge that the overall objective $\mathcal{L}(\theta)=\mathcal{L}(y, s_{\theta}(x))$ is non-convex w.r.t parameter set **$\theta$** for DNNs, even adding $l_2$ regularizer on $\theta$.
>
> **(3) The results of the prediction head do matter for the optimization of the whole network.**
>
> Even though the full network is non-convex, the last-layer prediction head forms a convex subproblem. Its smoothness constant directly modulates the gradient back-propagated into feature extractor $s_\theta(\cdot)$. Formally, for $p_\theta (x)=\text{Head}(s_\theta (x))$, the gradient w.r.t. feature parameters satisfies: $\nabla\_{\theta}\mathcal{L}=J_{\theta\|s}^\top \cdot \nabla_{s} \mathcal{L}\_{\text{Head}}$ , where $J_{\theta|s}$ is the Jacobian of $s_\theta(x)$ w.r.t. $\theta$. Consequently, the gradient magnitude can be upper-bounded by
>
> $
> \|\|\nabla_{\theta}\mathcal{L}\|\|\leq \|\|J_{\theta|s}^\top\|\|\cdot \|\|\nabla_s\mathcal{L}_{\text{Head}}\|\| \leq \|\|J\_{\theta|s}^\top\|\|\cdot L\_{\text{Head}} \cdot \|\| s - s^*\|\|,
> $
>
> where the smoothness of prediction head can scale and shape the gradient signal received by all earlier layers.
>
> **(4) Softmax Loss induces more favorable gradient properties for optimization compared to WOP objectives.**
> Hence, under the setting of identical backbone and architectural components, our analysis still implies that Softmax loss yields more favorable gradient properties than WOP surrogates. Such results of optimization properties are consistent with the empirical results observed in our experiments.
>
> Consequently, a more detailed explanation in our manuscript would be beneficial for clarification, which we promise to update a revision within the rebuttal period.
>
> [R1] Blondel et al., “Learning with Fenchel-Young Losses”, JMLR 2020

---

> > ### Comment · Reviewer_wSds · 2025-11-25
> >
> > I appreciate the authors’ timely response. I believe that the clarification of model’s convexity and explanation of the simplified analysis like ignoring the nonconvex components would further strengthen the manuscript. I keep my score to recommend acceptance.

---

> > > ### Author Response · Authors · 2025-11-26
> > >
> > > Thank you very much for your positive follow-up! Your feedback has helped us improve the clarity of presentation, and we are grateful for your thoughtful assessment. We will explicitly clarify this point and strengthen the explanation of why our simplified analysis is theoretically meaningful in the revised version of manuscript.

---

### Official Review · Reviewer_XpJH · 2025-10-31

**Soundness:** 3
**Presentation:** 2
**Contribution:** 3
**Rating:** 6
**Confidence:** 2

**Summary:**

This paper, offers a detailed theoretical study of the Softmax loss and related methods used in classification and ranking tasks. The main goal is to understand, in a unified way, how different versions of Softmax — such as Sparsemax, α-Entmax, Rankmax, and approximate methods behave both statistically and computationally. The authors use the Fenchel–Young (F–Y) framework, a convex-analytic approach, to describe all these losses under a common theory. They show the following:
1. The consistency properties of different F-Y surrogates
2. Analyze the gradient dynamics of Softmax-family losses, characterizing their convergence behaviors
3. Comparitive analysis of per epoch computational complexity
4. Complement with extensive experiments

**Strengths:**

1. The paper nicely brings together many known results on Softmax, Sparsemax, and their approximations under one unifiied theoretical framework
2. The analysis showing how Jacobian spectral norms explain convergence differences between losses is insightful

**Weaknesses:**

1. Can we interpret these Fenchel–Young losses as mirror descent updates, since each loss defines its own geometry?
2. There is a bunch of qualitative analysis but no explicit rates of convergence? Do they directly follow?
3. I am not clear on prior related works on the fenchel-young framework. Can the authors add that too if it is relevant?
4. I don’t really understand what the final takeaway of the unified analysis is. Maybe it is useful to emphasize that too. I see the tables in the beginning, but it might be useful to add that the abbreviations mean in that for people not in the area.

**Questions:**

See weakness!

---

> ### Author Response · Authors · 2025-11-20
>
> **We appreciate reviewer XpJH for your constructive and valuable comments.**
>
> We are glad that the reviewer praises the **“insightful”** analysis of Jacobian spectral norms and convergence behavior. Below we hope to address each of your concerns in detail.
>
> **W1: Connection between Fenchel–Young losses and Mirror Descent.**
>
> Thank you for raising this important theoretical perspective! F–Y losses and Mirror Descent (MD) are indeed closely connected, which lies in their shared geometry and optimality conditions. As mentioned in our paper, an F–Y loss is generated from a convex potential $\Omega$, whose Fenchel conjugate $\Omega^\*$ defines the model’s prediction rule $\hat{y} = \nabla\Omega^\*(\theta)$. Besides, $\Omega$ also induces a Bregman divergence
>
> $D_{\Omega}(u,v) = \Omega(u) - \Omega(v) - \langle \nabla \Omega(v), u-v \rangle,$
>
> which coincides with the geometric design utilized by MD through updates with
>
> $x_{t+1} = \arg\min_x [ \langle \nabla f(x_t), x \rangle + \frac{1}{\eta} D_{\Omega}(x, x_t) ] $.
>
> Therefore, the two frameworks share the same dual optimality condition induced by $\Omega$.
>
> The main difference lies in their update dynamics: MD optimizes a task objective $f(x)$ and converges to a stationary point where $\nabla f(x^\*) = 0$. Its dual variables are always linked with primal $x$ through the mirror map $\theta = \nabla\Omega(x)$ and its inverse $x = \nabla\Omega^\*(\theta)$. Thus MD naturally satisfies $x^\* = \nabla\Omega^\*(\theta^\*)$ at convergence. However, F–Y losses enforce the same condition in a supervised manner: minimizing an F–Y loss yields $\nabla_{\theta} L_{\Omega}(\theta; y) = \nabla\Omega^{\*}(\theta) - y,$ which drives the model to satisfy $\hat{y}=\nabla\Omega^{\*}(\theta)=y$. In this sense, an F–Y loss can be viewed as a supervised imitation of MD’s mirror-space optimal behavior by directly penalizing deviations of MD's optimal.
>
> Different choices of $\Omega$ correspond to different geometries and members of the softmax-family: Softmax <- Shannon entropy, Sparsemax <- Euclidean, and $\alpha$-Entmax <- Tsallis entropy. We have added a short clarification of this relationship in the Appendix of the revised manuscript.
>
> **W2: Explicit convergence rates**
>
> Thank you for raising this point out. We would like to clarify that the proposed theoretical results are sufficient to provide explicit convergence rate coefficient $\rho$ (with $\sim O(\rho^t)$) for each loss. Unlike classical convergence-rate studies, the differences between surrogate losses affect only the constants rather than order of convergence rates, because all losses share similar smoothness and strong-convexity structure once $\ell_2$​-regularization is applied. Consequently, all losses achieve similar asymptotic linear rate, and what meaningfully distinguishes them is the magnitude of the smoothness constants, which is directly governed by the Jacobian spectral norms. Therefore, the main text emphasizes how spectral-norm differences impact these constants, and we summarize the resulting ordering of convergence-rate coefficients in Eq. (7).
>
> **W3: Related work on the Fenchel–Young framework**
>
> We are glad to provide a more detailed discussion of the F–Y framework, as we believe this theoretical topic is valuable for readers who may not be familiar with the underlying convex-analytic viewpoint. Although Section 2.4 and Appendix A.2 of the paper review the basic definition of F–Y losses, their connection to Fenchel conjugacy, and their relationship to Bregman divergences, we agree that a clearer introduction of all prior work will further improve the overall presentation.
>
> Before the introduction of F–Y losses, the design of machine learning losses was largely heuristic. The F–Y framework, however, provides a principled and constructive alternative: starting from a single convex regularization function $\Omega$, one jointly derives both the prediction map $\hat{y} = \nabla\Omega^\*(\theta)$ and a convex, differentiable loss $L_\Omega$ ​. Once $\Omega$ is specified, the loss follows automatically, and its gradient is guaranteed to take the residual form $\hat{y} – y$. This framework unifies the design of activation function and provides a principled way to construct appropriate surrogate losses for complex or sparse output spaces, while ensuring the consistency properties of the prediction task (as proven in our paper).
>
> Furthermore, as discussed in our response to your first question, an F–Y loss can also be interpreted as a supervised counterpart of MD optimal condition. Minimizing an F–Y loss drives the dual variable $\theta$ to satisfy $\hat{y} = \nabla \Omega^\*(\theta) = y$. Thus, F–Y losses provide not only a constructive surrogate loss framework, but also a geometrically aligned objective induced by $\Omega$.

---

> > ### Author Response · Authors · 2025-11-20
> >
> > **Final Takeaways.**
> >
> > We agree that our principled analysis could better highlight its main takeaway. We have revised the manuscript to make the conclusions more explicit. In particular, based on Tables 1–3 in the paper, the unified takeaway can be concisely summarized as follows:
> >
> > **(1) When the number of classes $C$ is moderate and exact methods are feasible:**
> > Softmax provides SOP and smaller Jacobian spectral norm (higher smoothness), which corresponds to better-conditioned optimization and faster convergence compared with its sparse variants and should be tested in priority.
> >
> > **(2) When $C$ is extremely large and approximate methods are required:**
> > The choice among approximate softmax surrogates can be guided by the bias–variance decomposition. These results quantify how each approximation deviates from the exact Softmax risk, which allows us to identify the method with the smallest total deviation, and also assess whether bias can be further reduced, such as increasing the sample size $k$ or adapting the proposal distribution $Q$.

---

### Note · Program_Chairs · 2026-01-17
**Submission Desk Rejected by Program Chairs**

The following references in this submission do not refer to real documents and/or have major errors in bibliographic information:

 Pradeep Ravikumar, Alekh Agarwal, and Martin J Wainwright. Ndcg-consistent ranking surrogates. In Proceedings of the 28th International Conference on Machine Learning (ICML), pp. 641-648, 2011.